# Hierarchical Concept-based Interpretable Models

**Oscar Hill**
University of Cambridge
ogh22@cam.ac.uk

**Mateo Espinosa Zarlenga**
University of Oxford
trin4380@ox.ac.uk

**Mateja Jamnik**
University of Cambridge
mj201@cam.ac.uk

## Abstract

Modern deep neural networks remain challenging to interpret due to the opacity of their latent representations, impeding model understanding, debugging, and debiasing. Concept Embedding Models (CEMs) address this by mapping inputs to human-interpretable concept representations from which tasks can be predicted. Yet, CEMs fail to represent inter-concept relationships and require concept annotations at different granularities during training, limiting their applicability. In this paper, we introduce *Hierarchical Concept Embedding Models* (HiCEMs), a new family of CEMs that explicitly model concept relationships through hierarchical structures. To enable HiCEMs in real-world settings, we propose *Concept Splitting*, a method for automatically discovering finer-grained sub-concepts from a pretrained CEM's embedding space without requiring additional annotations. This allows HiCEMs to generate fine-grained explanations from limited concept labels, reducing annotation burdens. Our evaluation across multiple datasets, including a user study and experiments on *PseudoKitchens*, a newly proposed concept-based dataset of 3D kitchen renders, demonstrates that (1) Concept Splitting discovers human-interpretable sub-concepts absent during training that can be used to train highly accurate HiCEMs, and (2) HiCEMs enable powerful test-time concept interventions at different granularities, leading to improved task accuracy.

## 1 Introduction

State-of-the-art Deep Neural Networks (DNNs) can achieve very high task accuracies but fail to explain their reasoning in human-understandable terms (Barredo Arrieta et al., 2020). Concept Embedding Models (CEMs) (Espinosa Zarlenga et al., 2022) address this limitation by learning to predict a set of human-understandable *concept* representations provided at training time (e.g., "size" or "colour"), and then using these concept representations (or *embeddings*) for learning to predict a downstream task. Within this framework, a CEM's concept predictions serve as an explanation for its downstream task prediction. However, CEMs cannot model relationships between concepts, treating all concepts as independent entities from each other, leading to their representations failing to capture known inter-concept relationships (Raman et al., 2024). This is problematic because real-world concepts are often interrelated, and human cognition inherently utilises such relationships for reasoning (McClelland & Rogers, 2003). Additionally, CEMs require expensive concept annotations at training time to learn their embeddings (Espinosa Zarlenga et al., 2022), limiting their usability.

While numerous researchers have explored concept discovery (Yuksekgonul et al., 2023; Oikarinen et al., 2023; Rao et al., 2024), they typically overlook the hierarchical relationships between discovered concepts. Additionally, these methods rarely support human-in-the-loop refinement, where expert interventions, such as corrections to concept predictions at test time, could improve model performance. These gaps limit their applicability to real-world scenarios where hierarchical concept structures and iterative human feedback are critical.

In this paper, we show that CEMs capture sub-concepts not provided during training as part of their embedding spaces. For example, a CEM trained with the concept "contains vegetables" may encode subspaces corresponding to finer-grained sub-concepts like "contains onions" and "contains carrots" within its embedding manifold. To exploit this structure, we propose (1) *Concept Splitting* (Figure 1), a method for discovering sub-concepts from a CEM's concept embeddings using sparse autoencoders

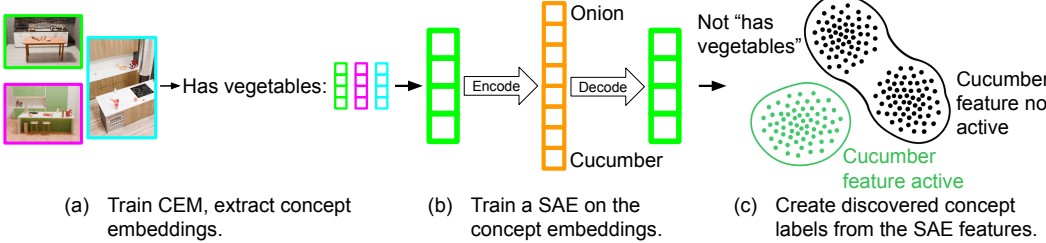

(a) Train CEM, extract concept embeddings.

(b) Train a SAE on the concept embeddings.

(c) Create discovered concept labels from the SAE features.

Figure 1: **Concept Splitting**. (a) Train a CEM and calculate concept embeddings. (b) Train SAEs on the embeddings (the image depicts a single embedding set). (c) Create concept labels. The green points are marked as having the new concept, and the black points (where the parent concept is not active or where the SAE feature is not active) are marked as not having the new concept.

(Bricken et al., 2023), and (2) *Hierarchical CEM* (HiCEM, Figures 2 and 3), a model designed to support hierarchical concept relationships like those discovered by Concept Splitting. Together, Concept Splitting and HiCEMs significantly reduce annotation costs by requiring only coarse, high-level concept labels during training while automatically discovering more granular sub-concepts. The coarse concept labels could be obtained cheaply by using LLMs (Oikarinen et al., 2023; Yang et al., 2023) or by grouping semantically related classes. Our evaluation across several datasets, including a user study, strongly suggests that Concept Splitting can discover human-interpretable concepts that HiCEMs can then utilise to construct more fine-grained explanations for their predictions without compromising predictive or intervention performance. Hence, our contributions are:

- We introduce **Concept Splitting**, a method to discover sub-concepts in a CEM's embeddings. This reduces the need for exhaustive concept annotations and enhances the granularity of explanations.
- We propose **HiCEMs**, a family of inherently interpretable concept-based models that capture hierarchical concept relationships and support human interventions at multiple hierarchy levels.
- We introduce **PseudoKitchens**, a synthetic dataset of photorealistic 3D kitchen renders with perfect ground-truth concept annotations and precise spatial localisation.
- We demonstrate, through empirical, qualitative, and quantitative experiments, including a user study, that HiCEMs trained via Concept Splitting can accurately discover interpretable sub-concepts that were absent during training. Moreover, our experiments show that HiCEMs trained with Concept Splitting achieve competitive task accuracies and are receptive to test-time concept interventions at different granularity levels.

## 2 BACKGROUND AND RELATED WORK

**Concept learning** Concept-based methods aim to explain a model's predictions using human-understandable concepts (e.g., "colour" or "size") (Bau et al., 2017; Fong & Vedaldi, 2018; Kim et al., 2018). Some methods, for example, Concept Bottleneck Models (CBMs) (Koh et al., 2020), explicitly incorporate concepts in their architecture, leading to inherently interpretable models that provide concept-based explanations. These methods typically require a concept-annotated training set and may suffer from suboptimal predictive performance due to conflicting training objectives (Espinosa Zarlenga et al., 2022). Furthermore, they often do not consider the relationships between concepts; instead, they treat all concepts as independent variables (Havasi et al., 2022). Other methods attempt to construct post-hoc concept-based explanations, instead of producing inherently interpretable models. For example, Automatic Concept-based Explanations (ACE) (Ghorbani et al., 2019) clusters a DNN's latent space to discover relevant concepts. In contrast to ACE, we (1) use the discovered concepts to construct inherently interpretable models, (2) exploit the relationship between existing and discovered concepts, and (3) demonstrate effective discovered concept interventions.

**Concept Embedding Models** Concept Embedding Models (CEMs) (Espinosa Zarlenga et al., 2022) provide concept-based explanations while achieving higher task accuracies than CBMs. CEMs, and their variants (Kim et al., 2023; Espinosa Zarlenga et al., 2024; Xu et al., 2024; Espinosa Zarlenga et al., 2025), improve task accuracy by representing concepts using high-dimensional supervised

vectors, or *embeddings*. For each concept $c_i$, a CEM learns two embeddings: one for when it is active (a "positive" embedding, $\hat{\mathbf{c}}_i^+$), and another for when it is inactive (a "negative" embedding, $\hat{\mathbf{c}}_i^-$). Each concept embedding is aligned to its corresponding ground truth concept through a scoring function $s$, which learns to assign an activation probability $\hat{p}_i$ to each concept $c_i$. These probabilities are used to output an embedding $\hat{\mathbf{c}}_i$ for each concept $c_i$ via a mixture of positive and negative embeddings weighted by the predicted probability (i.e., $\hat{\mathbf{c}}_i = \hat{p}_i\hat{\mathbf{c}}_i^+ + (1 - \hat{p}_i)\hat{\mathbf{c}}_i^-$). Finally, the mixed concept embeddings are concatenated into a bottleneck $\hat{\mathbf{c}}$ and passed to a label predictor $f(\hat{\mathbf{c}})$ (usually a linear layer). This predictor outputs a downstream task prediction $\hat{y}$.

An important property of CEMs is that they support *concept interventions*: at test time, an expert can correct mispredicted concepts, allowing the model to update its downstream label prediction based on these corrections. Interventions can be performed by fixing a concept's embedding to the embedding that is semantically aligned with the ground truth concept label (e.g., setting $\hat{\mathbf{c}}_i := \hat{\mathbf{c}}_i^+$ if concept $c_i$ is determined to be present in $x$). However, a major limitation of CEMs is that they fail to capture known inter-concept relationships (Raman et al., 2024), deviating from how humans tend to reason about a task hierarchically (McClelland & Rogers, 2003). Moreover, lacking a mechanism for representing hierarchical inter-concept relationships means that CEMs need a large number of concept labels in their training datasets to capture different concept granularities. Our work addresses these issues by exploiting a pre-trained CEM's embedding space to discover sub-concepts that were not provided during training, and using them to train HiCEMs that can exploit sub-concept relationships.

**Concept discovery**  Several methods (Bricken et al., 2023; Huang et al., 2022; Huben et al., 2024; O'Mahony et al., 2023; Rao et al., 2024; Vielhaben et al., 2023; Yuksekgonul et al., 2023; Zhang et al., 2021) aim to identify human-interpretable concepts that can help explain how a model makes its predictions. Some methods try to discover concepts encoded by a model's neurons (Fel et al., 2023; Graziani et al., 2023; Oikarinen & Weng, 2023; Panousis & Chatzis, 2023), some ask large language models to suggest relevant concepts (Oikarinen et al., 2023; Yang et al., 2023), and some look for meaningful features of inputs (Ghorbani et al., 2019). A few approaches attempt to assign concepts to individual neurons, however this can be problematic, as neurons often represent complex, uninterpretable features (Elhage et al., 2022). However, most of these works typically fail to evaluate interventions on discovered concepts or to consider the relationships between discovered concepts. For example, hierarchical relationships among concepts imply that some sub-concepts (e.g., "contains onions" and "contains carrots") can only exist in the presence of another parent concept (e.g., "contains vegetables"). Therefore, here we present a method for discovering such sub-concepts and propose an inherently interpretable architecture that explicitly represents these sub-concept relationships. Furthermore, we demonstrate that interventions on discovered sub-concepts are effective.

**Modelling concept relationships**  Most early concept-based models unrealistically assumed inter-concept independence. More recent works model inter-concept relationships through autoregressive architectures (Havasi et al., 2022), causal graphs (Dominici et al., 2024), probabilistic methods (Vandenhirtz et al., 2024; Xu et al., 2024), or intervention-specific mechanisms (Singhi et al., 2024). However, these capture interactions without explicitly organising concepts into hierarchies or discovering sub-concepts. A related direction by Panousis et al. (2024) models hierarchical concepts but uses a spatially-grounded approach (high-level for whole images, low-level for patches) dependent on vision-language models and textual concept descriptions. In contrast, we discover hierarchies by decomposing concept embeddings, independent of spatial structure or vision-language constraints. To enable this, we introduce *Concept Splitting*, a method to automatically discover latent sub-concepts from a model trained with only high-level labels, and *HiCEMs*, an architecture that explicitly models these hierarchical relationships to enable fine-grained explanations and multi-level interventions without exhaustive annotations.

## 3  CONCEPT SPLITTING

Previous work has found that Sparse AutoEncoders (SAEs) can uncover interpretable concepts in the representation spaces of neural networks (Bricken et al., 2023; Bussmann et al., 2024). SAEs are a type of autoencoder trained to reconstruct their input while enforcing a sparsity constraint on a high-dimensional latent representation, effectively learning a dictionary of (hopefully interpretable) features. We apply this idea in CEMs to discover sub-concepts, and then we train HiCEMs (Section 4)

that use the discovered concepts to provide finer-grained explanations. Specifically, we use BatchTopK sparse autoencoders (Bussmann et al., 2024), which keep the top activations across a batch. While we focus on using SAEs to discover sub-concepts, they are not the only option, and Appendix A discusses an alternative approach using clustering to find mutually exclusive sub-concepts.

Our approach, which we name *Concept Splitting* (Figure 1), takes as input a trained CEM $M$. First, we run $M$ on a concept-annotated training set $\mathcal{D} = \{(\mathbf{x}^{(j)}, \mathbf{c}^{(j)}, y^{(j)})\}_{j=1}^{N}$, storing the concept embedding vectors and concept predictions $\{\hat{\mathbf{c}}_i, \hat{p}_i\}$ for each sample in $\mathcal{D}$ and for each concept $c_i$ we want to split (Figure 1(a)). For simplicity, we describe how to split a single concept $c_i$. However, this operation can be performed for all training concepts.

Next, let $E_i$ be the set of embedding vectors for $c_i$, and let $E_i^{\mathbf{true}}$ and $E_i^{\mathbf{false}}$ be the subsets of embeddings in $E_i$ where $c_i$ was predicted by $M$ to be present (i.e., $\hat{p}_i > 0.5$) or absent, respectively. We partition the embedding vectors using $M$'s concept predictions for $c_i$, as these predictions tell us whether the dominant component in the mixture is a positive embedding or a negative embedding.

Here, we want to discover sub-concepts of $c_i$, where we consider the sub-concepts of $c_i$ to be groups of concepts that are either only active when $c_i$ is (positive sub-concepts), or only active when $c_i$ is not (negative sub-concepts). To this end, we train SAEs on $E_i^{\mathbf{true}}$ and $E_i^{\mathbf{false}}$ separately (Figure 1(b)). Using the SAE trained on $E_i^{\mathbf{true}}$, we discover sub-concepts that are present when $c_i$ is also present, and using the SAE trained on $E_i^{\mathbf{false}}$ we discover sub-concepts that are present when $c_i$ is not.

Once we have trained a SAE on an embedding set, we create new concept labels using the features learned by the SAE (Figure 1(c)). Bussmann et al. (2024) describe how to calculate a threshold for determining when a feature is active during inference. Every feature can be treated as a discovered sub-concept. The examples that activate the feature are marked as having the new sub-concept, and the examples that do not activate the feature are marked as not having the new sub-concept.

Once Concept Splitting has been performed, we can interpret a discovered sub-concept using proto-types, which provide training examples that strongly activate the concept. This approach, similar to that used by previous works (Alvarez Melis & Jaakkola, 2018; Espinosa Zarlenga et al., 2023; Yeh et al., 2020), enables experts to assign potential semantics to discovered concepts.

## 4 Hierarchical CEMs

We introduce the HiCEM architecture (Figure 2), which explicitly models hierarchical relationships between concepts. For simplicity, we focus on two-level hierarchies, however our architecture could be extended to support deeper hierarchies.

### 4.1 Architecture

Like in CEMs, for each top-level concept, a HiCEM learns a mixture of two embeddings with semantics representing the concept's activity. In HiCEM, each top-level concept $c_i$ is represented with the embeddings $\hat{\mathbf{c}}_i^{+}, \hat{\mathbf{c}}_i^{-} \in \mathbb{R}^m$. Here, $\hat{\mathbf{c}}_i^{+}$ represents $c_i$'s active state, and $\hat{\mathbf{c}}_i^{-}$ represents its inactive state. In contrast to CEMs however, we also want $\hat{\mathbf{c}}_i^{+}$ and $\hat{\mathbf{c}}_i^{-}$ to contain information about $c_i$'s positive and negative sub-concepts, respectively. To achieve this, a backbone network $\psi(\mathbf{x})$ (e.g., a pre-trained ResNet model) produces a latent representation $\mathbf{h} \in \mathbb{R}^{n_{\text{hidden}}}$ which is the input to top-level embedding generators $\phi_i^{+}$ and $\phi_i^{-}$. These top-level embedding generators produce intermediate embeddings $\hat{\mathbf{c}}_i^{+\prime} = \phi_i^{+}(\mathbf{h}), \hat{\mathbf{c}}_i^{-\prime} = \phi_i^{-}(\mathbf{h}) \in \mathbb{R}^m$. Following the work of Espinosa Zarlenga et al. (2022), we implement the top-level embedding generators as single fully connected layers.

To produce final concept embeddings that contain information about sub-concepts, the embeddings $\hat{\mathbf{c}}_i^{+\prime}$ and $\hat{\mathbf{c}}_i^{-\prime}$ are passed through a positive and a negative *sub-concepts module*, respectively. The positive sub-concepts module, which we describe in further detail below, is responsible for learning the positive sub-concepts of $c_i$. It outputs the positive concept embedding for $c_i$, $\hat{\mathbf{c}}_i^{+}$, as well as the probability of its most likely positive sub-concept, $\hat{p}_i^{+}$. Similarly, the negative sub-concepts module outputs the negative concept embedding for $c_i$, $\hat{\mathbf{c}}_i^{-}$, as well as the probability of its most likely negative sub-concept, $\hat{p}_i^{-}$. If concept $c_i$ has no positive sub-concepts (i.e., concept $c_i$ is a leaf node in the hierarchy), then we take $\hat{\mathbf{c}}_i^{+} = \hat{\mathbf{c}}_i^{+\prime}$ and $\hat{p}_i^{+} = s(\hat{\mathbf{c}}_i^{+})$, where $s$ is a shared scoring function that calculates concept probabilities from concept embeddings. We proceed analogously in the absence of negative

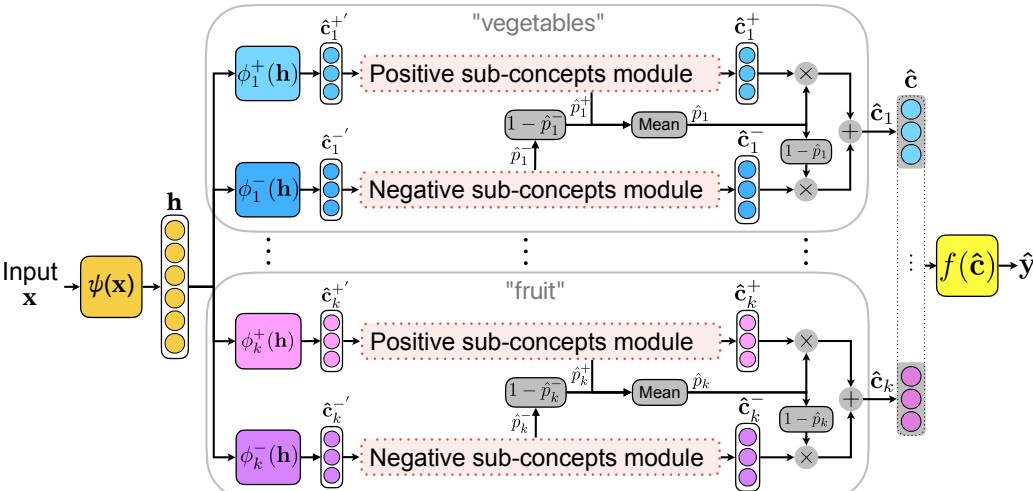

Figure 2: **Hierarchical CEM:** as in a CEM, from a latent code $\mathbf{h}$, we learn two embeddings per concept ($\hat{\mathbf{c}}_i^{+\prime}$ and $\hat{\mathbf{c}}_i^{-\prime}$). These embeddings are then passed through sub-concepts modules (Figure 3), which produce new embeddings ($\hat{\mathbf{c}}_i^+$ and $\hat{\mathbf{c}}_i^-$) that include information about sub-concepts. The sub-concepts modules also output the most likely sub-concept probabilities, which are used to calculate top-level concept probabilities. These probabilities are used to output an embedding for each concept via a weighted mixture of positive and negative embeddings.

sub-concepts. $\hat{p}_i^+$ can be taken as an estimate for the probability of the top-level concept $c_i$, because $c_i$ can only be present if one of its positive sub-concepts is. Similarly, the complement of $\hat{p}_i^-$ can be taken as another estimate for the probability of $c_i$. The predicted probability of concept $c_i$, $\hat{p}_i$, is calculated as the average of $\hat{p}_i^+$ and the complement of $\hat{p}_i^-$: $\hat{p}_i = \frac{1}{2}(\hat{p}_i^+ + 1 - \hat{p}_i^-)$. As in CEMs, the final concept embedding $\hat{\mathbf{c}}_i$ for $c_i$ is calculated as a weighted mixture of $\hat{\mathbf{c}}_i^+$ and $\hat{\mathbf{c}}_i^-$: $\hat{\mathbf{c}}_i = \hat{p}_i \hat{\mathbf{c}}_i^+ + (1 - \hat{p}_i)\hat{\mathbf{c}}_i^-$.

Like in previous embedding-based concept models (Espinosa Zarlenga et al., 2022; 2023; Xu et al., 2024), before making a task prediction, all $k$ mixed concept embeddings are concatenated, resulting in a bottleneck $g(\mathbf{x}) = \hat{\mathbf{c}}$ with $k \cdot m$ units. This is put through a label predictor $f$ to get a downstream task label. Following previous work (Koh et al., 2020; Espinosa Zarlenga et al., 2022), we use an interpretable label predictor $f$ parametrised by a simple linear layer.

As in previous concept-based models, HiCEMs provide a concept-based explanation for the predicted downstream task label through predicted concept probabilities $\hat{\mathbf{p}}(\mathbf{x}) \triangleq [\hat{p}_1, \ldots, \hat{p}_k]$. However, unlike previous architectures, HiCEMs explicitly model the relationship between concepts and sub-concepts: a concept's positive embedding contains information about its positive sub-concepts, and a concept's negative embedding includes information on its negative sub-concepts.

## 4.2 SUB-CONCEPTS MODULES

We describe a positive sub-concepts module, as illustrated in Figure 3, but negative sub-concepts modules operate in exactly the same way. Inside the positive sub-concepts module for concept $c_k$, sub-concept embedding generators $\phi_{kj}^+$ produce embeddings for each of $c_k$'s positive sub-concepts: $\hat{\mathbf{c}}_{kj}^+ = \phi_{kj}^+(\hat{\mathbf{c}}_k^{+\prime})$. Like the top-level embedding generators, these are implemented as single fully connected layers. Similarly to CEMs (Espinosa Zarlenga et al., 2022), these embeddings are aligned with ground-truth sub-concept $c_{kj}^+$ via a learnable and differentiable scoring function $s : \mathbb{R}^m \to [0, 1]$, trained to predict the probability $\hat{p}_{kj}^+ \triangleq s(\hat{\mathbf{c}}_{kj}^+) = \sigma(W_s \hat{\mathbf{c}}_{kj}^+ + \mathbf{b}_s)$ of sub-concept $c_{kj}^+$ being active from its sub-concept embedding. As in CEMs, the scoring function is shared across all sub-concepts. $c_k$'s positive embedding, $\hat{\mathbf{c}}_k^+$, is constructed as a weighted mixture of all the $n_k^+$ positive sub-concept embeddings: $\hat{\mathbf{c}}_k^+ \triangleq \sum_{j=1}^{n_k^+} \hat{p}_{kj}^+ \hat{\mathbf{c}}_{kj}^+$.

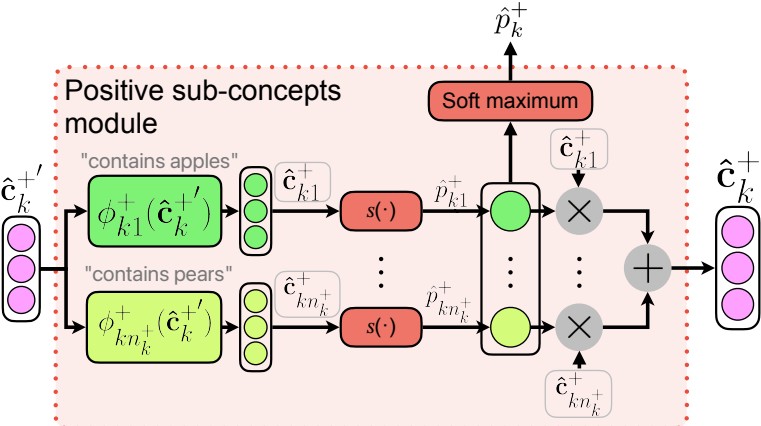

Figure 3: **Positive sub-concepts module:** positive sub-concept embedding generators $\phi_{kj}^+$ (illustrated for "contains apples" and "contains pears") produce sub-concept embeddings from the preliminary parent concept embedding $\hat{\mathbf{c}}_k^{+\prime}$. A shared scoring function $s(\cdot)$ predicts probabilities for each sub-concept. The positive parent concept embedding $\hat{\mathbf{c}}_k^+$ is computed as a weighted mixture of sub-concept embeddings, whilst an estimate for the parent concept probability $\hat{p}_k^+$ is obtained via a differentiable soft maximum operation over the sub-concept probabilities.

The highest positive sub-concept probability $\hat{p}_k^+$, which is an estimate for the probability of $c_k$, is calculated in a differentiable way as $\hat{p}_k^+ = \sum_{j=1}^{n_k^+} \texttt{softmax}(\alpha \cdot \hat{\mathbf{p}}_k^+ - \beta)_j \cdot \hat{p}_{kj}^+$, where $(\hat{\mathbf{p}}_k^+)_j = \hat{p}_{kj}^+$ and the constants $\alpha$ and $\beta$ scale the input of the $\texttt{softmax}$ so that its output is strongly weighted towards the largest $\hat{p}_{kj}^+$. In practice, we use $\alpha = 200, \beta = 100$ to scale the probabilities from $[0, 1]$ to $[-100, 100]$, so that the calculated highest sub-concept probability is very close to the true maximum of the probabilities.

## 4.3 TRAINING

HiCEMs are trained by jointly minimising a weighted sum of the cross-entropy loss on both task prediction and concept predictions: $\mathcal{L} \triangleq \mathbb{E}_{(\mathbf{x}, y, \mathbf{c})} \left[ \mathcal{L}_{\text{task}}(y, f(g(\mathbf{x}))) + \lambda \mathcal{L}_{\text{CrossEntr}}(\mathbf{c}, \hat{\mathbf{p}}(\mathbf{x})) \right]$. Here, the hyperparameter $\lambda \in \mathbb{R}^+$ controls the relative importance of concept and task accuracy.

When training HiCEMs, we use the RandInt regularisation strategy proposed by Espinosa Zarlenga et al. (2022) to improve intervention effectiveness. That is, during training, we randomly perform independent concept interventions with probability $p_{\text{int}}$.

## 4.4 CONCEPT INTERVENTIONS

HiCEMs support interventions on both top-level concepts and sub-concepts in a natural way. Similarly to CEMs (Espinosa Zarlenga et al., 2022), to intervene on top-level concept $c_i$, the embedding $\hat{\mathbf{c}}_i$ is updated by replacing it with the output embedding semantically aligned with the concept's ground truth label. This intervention ensures that $\hat{\mathbf{c}}_i$ contains information about the relevant sub-concepts.

Intervening on sub-concept $c_{ij}^+$ differs depending on whether the human expert decides $c_{ij}^+$ is or is not actually present in the example. (While we focus on positive sub-concepts here, the analogous process applies to negative sub-concepts.) If the human expert determines that $c_{ij}^+$ is not present, then the model's predicted probability $\hat{p}_{ij}^+$ is simply set to zero. On the other hand, if the expert identifies $c_{ij}^+$ as present, then $\hat{p}_{ij}^+$ is set to one, and $\hat{p}_{ij'}^+$ is set to zero for all $j' \neq j$ (this means $\hat{\mathbf{c}}_i^+ := \hat{\mathbf{c}}_{ij}^+$). Additionally, because the presence of $c_{ij}^+$ implies that the top-level concept $c_i$ is present, we also intervene on $c_i$. Therefore this operation results in the update $\hat{\mathbf{c}}_i := \hat{\mathbf{c}}_i^+ := \hat{\mathbf{c}}_{ij}^+$.

## 5 EXPERIMENTS

We evaluate Concept Splitting and HiCEMs by exploring the following research questions:

**RQ1** Does Concept Splitting discover interpretable sub-concepts?

**RQ2** How do HiCEMs' task and provided (top-level) concept accuracies compare to those of the original CEM and other baselines?

**RQ3** Can a HiCEM's task accuracy be improved by intervening on discovered sub-concepts?

### 5.1 PSEUDOKITCHENS

To rigorously evaluate concept-based models, we introduce PseudoKitchens (Appendix B), a synthetic dataset of photorealistic 3D kitchen renders with perfect ground-truth concept annotations. Our approach provides complete control over scene generation and pixel-perfect labels for all concepts.

### 5.2 SETUP

**Datasets** We evaluate our methods across six diverse datasets: MNIST-ADD (based on LeCun et al. (2010)), a procedural SHAPES dataset (similar to Matthey et al. (2017)), Caltech-UCSD Birds-200-2011 (CUB) (Wah et al., 2011), Animals with Attributes 2 (AwA2) (Xian et al., 2019), our new PseudoKitchens dataset, and ImageNet (Russakovsky et al., 2015). These datasets span simple synthetic tasks, fine-grained visual classification, and large-scale recognition, providing a comprehensive testbed. Full details on each dataset, including concept definitions and splits, are provided in Appendix C.

**Metrics** For each dataset, we run Concept Splitting on the provided concepts in an initial CEM, and then train a HiCEM with the provided top-level concepts and the discovered sub-concepts.

Following Espinosa Zarlenga et al. (2023), we evaluate discovered sub-concept accuracy and perform interventions by automatically pairing sub-concepts with a human-understandable "left-out" concept from a predefined "concept bank". This bank contains anticipated concepts excluded during initial CEM training (e.g., "the first digit is 6" in the MNIST-ADD dataset), with each concept pre-associated with the parent concept whose sub-concept it might represent. To align discovered sub-concepts with the bank, we compute the area under the receiver operating characteristic curve (ROC-AUC) scores between the discovered sub-concept labels and their potential parent-concept-associated matches in the bank. Each concept in the bank is assigned to the sub-concept with the highest ROC-AUC score, as long as that score is greater than 0.7. Discovered sub-concepts that are not matched with a concept bank concept are not included in the evaluation. We do not have ground-truth concept annotations for ImageNet (Russakovsky et al., 2015), so we evaluate the discovered sub-concepts with a user study. Discovered sub-concepts that were not matched to the concept bank, or that were not selected for the user study in the case of ImageNet, are not included when we train our HiCEMs.

To answer **RQ1**, we present the results of our user study conducted with ImageNet, and for the other datasets, we report the average discovered sub-concept ROC-AUC of the HiCEM. To address **RQ2**, we report the task accuracy and the provided concept ROC-AUC of the initial CEM and the HiCEM. For **RQ3**, we measure the change in HiCEMs' task accuracies as concepts are intervened. All metrics in our evaluation, apart from those calculated on ImageNet, are computed on test datasets using three random seeds, from which we compute a mean and standard deviation. Because ImageNet is so large, we compute metrics using a single random seed. Whenever we measure concept accuracy, we use the mean concept ROC-AUC to avoid being misled by a majority-class classifier.

**Baselines** We compare HiCEMs with black box models, CEMs (Espinosa Zarlenga et al., 2022), CBMs (Koh et al., 2020), Label-free CBMs (*LF-CBMs*, (Oikarinen et al., 2023)), Post-hoc CBMs (*PCBMs*, (Yuksekgonul et al., 2023)), and PCBMs with residual connections (*PCBM-hs*). Our black box models have the same architecture as our CEMs, but without any concept supervision. To evaluate the usefulness of the discovered concept labels produced by Concept Splitting, in each of our runs we train a control HiCEM that has the same architecture as the HiCEM with discovered sub-concepts, but no sub-concept supervision ("*HiCEM w/o concept splitting*"). That is, the HiCEM is trained with access to top-level concept labels but without any labels or supervision for sub-concepts. We

Table 1: Mean ROC-AUC for discovered concepts. LF-CBMs were unable to discover concepts on the PseudoKitchens dataset. Sub-concepts discovered with Concept Splitting are predicted accurately.

|  | MNIST-ADD | SHAPES | CUB | AwA2 | PseudoKitchens |
|---|---|---|---|---|---|
| LF-CBM | – | $0.75_{\pm 0.00}$ | $0.77_{\pm 0.00}$ | $0.78_{\pm 0.00}$ | – |
| HiCEM w/o concept splitting (control) | $0.86_{\pm 0.01}$ | $0.88_{\pm 0.02}$ | $0.81_{\pm 0.00}$ | $0.73_{\pm 0.00}$ | $0.75_{\pm 0.06}$ |
| CEM + Concept Splitting (ours) | $\mathbf{0.93}_{\pm 0.01}$ | $\mathbf{0.93}_{\pm 0.01}$ | $\mathbf{0.85}_{\pm 0.01}$ | $\mathbf{0.88}_{\pm 0.01}$ | $\mathbf{0.88}_{\pm 0.01}$ |
| HiCEM + Concept Splitting (ours) | $\mathbf{0.93}_{\pm 0.01}$ | $\mathbf{0.93}_{\pm 0.01}$ | $\mathbf{0.85}_{\pm 0.01}$ | $\mathbf{0.88}_{\pm 0.01}$ | $\mathbf{0.88}_{\pm 0.00}$ |

Table 2: User study results. Users are far more likely to say sub-concepts generated by our method are examples of their parent concept than randomly chosen words are (first two rows). Users also agree that, in a lot of cases, the images labelled as having a discovered sub-concept are consistent with the automatically generated name of that sub-concept (second two rows).

|  | Yes | No | Not sure | They are the same |
|---|---|---|---|---|
| Sub-concept relationship (control) | 16 (3.8%) | 387 (91.3%) | 20 (4.7%) | 1 (0.2%) |
| Sub-concept relationship (experimental) | 241 (60.6%) | 109 (27.4%) | 19 (4.8%) | 29 (7.3%) |
| Image labels (control) | 4 (0.9%) | 424 (97.9%) | 5 (1.2%) | – |
| Image labels (experimental) | 244 (54.8%) | 172 (38.7%) | 29 (6.5%) | – |

match the "sub-concepts" in this baseline to our concept bank using sub-concept predictions on the training dataset. We also train a CEM with top-level concepts *and* discovered sub-concepts ("*CEM + Concept Splitting*") so we can compare this to the HiCEM. The CUB dataset is used by Oikarinen et al. (2023) to evaluate LF-CBMs, so we take their discovered concepts and manually match them with concepts for which we have ground-truth labels. For the other datasets, and for all the datasets with the PCBM baseline, we use the names of concepts for which we have ground-truth labels to create the models. For further details on how we train each of these baselines, including the architectures and hyperparameters used, see Appendices D and E. Due to the computational cost of training models on ImageNet, we do not include results for all baselines on it.

**User Study**    To evaluate the sub-concepts discovered on ImageNet, we ran a user study. We automatically named discovered sub-concepts with the CLIP vision-language model (Radford et al., 2021), using a method similar to previous work (Rao et al., 2024). We then asked users whether the discovered sub-concept names were semantically related to the name of their parent concept ("Is [sub-concept name] an example of [parent concept name]?"). As a control we picked words at random from the dictionary used to name the sub-concepts. We also asked users whether images labelled by Concept Splitting as having the discovered sub-concepts were consistent with the sub-concept names ("Does this image show [sub-concept name]?"), and used images selected at random from the dataset as a control. Full details are contained in Appendix F.

## 5.3    RESULTS

**Discovered concepts are human-interpretable and can be predicted accurately (RQ1, Tables 1 and 2).**    We report the accuracy of the discovered concept predictions made by our models using the ground truth labels of the corresponding human-interpretable concept bank concepts on the test datasets. For example, the meaning assigned to one of the concepts discovered on the MNIST-ADD dataset was "the top digit is 6", and therefore we compute the accuracy of this discovered concept with respect to the ground-truth labels of the concept "the top digit is 6" in our concept bank. In Appendix G, we show samples from the training dataset that were labelled as having this discovered concept, to demonstrate how it is straightforward to interpret it. Table 1 shows that the mean discovered sub-concept ROC-AUCs are high, exceeding 0.9 in some cases. The mean discovered concept ROC-AUC of HiCEMs is always higher than that of both LF-CBMs and HiCEMs trained without sub-concept supervision (HiCEM w/o concept splitting in Table 1), showing that the labels produced by Concept Splitting align the sub-concept activations in HiCEMs with human-interpretable concepts. CEMs with discovered sub-concepts and top-level concepts side by side have similar discovered concept accuracies to HiCEMs after Concept Splitting, however sub-concept interventions in HiCEMs can work better than in CEMs (Figure 4, discussed later).

Table 3: Task accuracies. The task accuracy of HiCEMs is competitive with all our baselines. * indicates reported accuracy, with a ResNet-50 backbone.

| | MNIST-ADD | SHAPES | CUB | AwA2 | PseudoKitchens | ImageNet |
|---|---|---|---|---|---|---|
| Black box (not interpretable) | $0.94_{\pm 0.00}$ | $0.89_{\pm 0.00}$ | $0.80_{\pm 0.00}$ | $0.98_{\pm 0.00}$ | $0.67_{\pm 0.01}$ | 0.77 |
| LF-CBM | – | $0.59_{\pm 0.01}$ | $\mathbf{0.80}_{\pm \mathbf{0.00}}$ | $0.94_{\pm 0.00}$ | – | 0.72* |
| PCBM | $0.16_{\pm 0.03}$ | $0.54_{\pm 0.01}$ | $0.65_{\pm 0.01}$ | $0.95_{\pm 0.00}$ | $0.16_{\pm 0.00}$ | – |
| PCBM-h | $0.53_{\pm 0.01}$ | $0.73_{\pm 0.00}$ | $0.73_{\pm 0.00}$ | $0.96_{\pm 0.00}$ | $0.52_{\pm 0.01}$ | – |
| CBM | $0.23_{\pm 0.01}$ | $0.78_{\pm 0.01}$ | $0.65_{\pm 0.00}$ | $0.97_{\pm 0.00}$ | $\mathbf{0.66}_{\pm \mathbf{0.01}}$ | 0.77 |
| CEM | $0.92_{\pm 0.01}$ | $\mathbf{0.89}_{\pm \mathbf{0.00}}$ | $0.76_{\pm 0.01}$ | $0.98_{\pm 0.00}$ | $0.66_{\pm 0.01}$ | $\mathbf{0.79}$ |
| HiCEM w/o concept splitting (control) | $\mathbf{0.93}_{\pm \mathbf{0.00}}$ | $0.88_{\pm 0.01}$ | $0.77_{\pm 0.01}$ | $0.98_{\pm 0.00}$ | $0.64_{\pm 0.01}$ | – |
| CEM + Concept Splitting (ours) | $\mathbf{0.93}_{\pm \mathbf{0.01}}$ | $0.87_{\pm 0.01}$ | $0.79_{\pm 0.01}$ | $0.98_{\pm 0.00}$ | $\mathbf{0.66}_{\pm \mathbf{0.02}}$ | – |
| HiCEM + Concept Splitting (ours) | $0.92_{\pm 0.00}$ | $0.87_{\pm 0.02}$ | $0.74_{\pm 0.01}$ | $\mathbf{0.98}_{\pm \mathbf{0.00}}$ | $0.65_{\pm 0.01}$ | 0.78 |

Table 4: Mean ROC-AUCs for provided concepts. HiCEMs are able to predict provided concepts just as well as CEMs.

| | MNIST-ADD | SHAPES | CUB | AwA2 | PseudoKitchens | ImageNet |
|---|---|---|---|---|---|---|
| CBM | $\mathbf{0.99}_{\pm \mathbf{0.00}}$ | $\mathbf{1.00}_{\pm \mathbf{0.00}}$ | $0.89_{\pm 0.00}$ | $\mathbf{1.00}_{\pm \mathbf{0.00}}$ | $0.91_{\pm 0.00}$ | 0.99 |
| CEM | $\mathbf{0.99}_{\pm \mathbf{0.00}}$ | $\mathbf{1.00}_{\pm \mathbf{0.00}}$ | $\mathbf{0.95}_{\pm \mathbf{0.00}}$ | $\mathbf{1.00}_{\pm \mathbf{0.00}}$ | $\mathbf{0.92}_{\pm \mathbf{0.00}}$ | $\mathbf{1.00}$ |
| HiCEM w/o concept splitting (control) | $\mathbf{0.99}_{\pm \mathbf{0.00}}$ | $\mathbf{1.00}_{\pm \mathbf{0.00}}$ | $0.93_{\pm 0.00}$ | $\mathbf{1.00}_{\pm \mathbf{0.00}}$ | $0.91_{\pm 0.00}$ | – |
| CEM + Concept Splitting (ours) | $\mathbf{0.99}_{\pm \mathbf{0.00}}$ | $\mathbf{1.00}_{\pm \mathbf{0.00}}$ | $\mathbf{0.95}_{\pm \mathbf{0.00}}$ | $\mathbf{1.00}_{\pm \mathbf{0.00}}$ | $\mathbf{0.92}_{\pm \mathbf{0.00}}$ | – |
| HiCEM + Concept Splitting (ours) | $\mathbf{0.99}_{\pm \mathbf{0.00}}$ | $\mathbf{1.00}_{\pm \mathbf{0.00}}$ | $0.93_{\pm 0.01}$ | $\mathbf{1.00}_{\pm \mathbf{0.00}}$ | $0.91_{\pm 0.00}$ | 0.99 |

Our user study on ImageNet (Table 2), conducted with 20 participants, shows that the sub-concepts discovered by our method are both semantically coherent and accurately labelled. When evaluating the names of sub-concepts generated by Concept Splitting, participants found them to be semantically related to their parent concepts **67.9%** of the time, a dramatic increase over the **4.0%** agreement rate for randomly chosen words in the control group. Furthermore, users confirmed that images labelled by our method were consistent with the discovered sub-concept name in **54.8%** of cases, far exceeding the **0.9%** agreement for the control. A Chi-Square test (where we discard "Not sure" and group "Yes" and "They are the same" together) confirms that both of these improvements over the control groups are statistically significant ($p < 0.01$).

To assess the robustness of Concept Splitting across different embedding representations, we further explore an idealised setting in Appendix H, where concept embeddings contain perfect one-hot information about sub-concepts. Here, Concept Splitting recovers the encoded sub-concepts almost perfectly. This suggests that our method can be applied to arbitrary concept embedding spaces as long as they encode sub-concept information: it is not dependent on the embedding structure of CEMs.

**HiCEMs have high task and provided concept accuracies (RQ2, Tables 3 and 4).** We measure the task and provided (top-level) concept accuracies of HiCEMs and our baselines. The results are in Tables 3 and 4. HiCEMs achieve both high task accuracy and high provided concept accuracy, compared to the baselines. In particular, the task and provided concept accuracies of HiCEMs are never more than 2% below those of CEMs, so running Concept Splitting and replacing a CEM with a HiCEM that supports more detailed explanations does not lead to a reduction in task or provided concept accuracy. Overall, the effect of Concept Splitting on the task and provided concept accuracy is insignificant, so the additional interpretability offered by the discovered sub-concepts does not come at the cost of accuracy.

**Intervening on sub-concepts identified through Concept Splitting can enhance task accuracy, with these interventions sometimes yielding even greater improvements in HiCEMs compared to CEMs. (RQ3).** We investigate how intervening on provided and discovered concepts affects task accuracy in HiCEMs and our relevant baselines. Figure 5 demonstrates that provided concept interventions perform equally well in HiCEMs as in CEMs.

As shown in Figure 4, intervening on discovered sub-concepts can lead to an increase in task accuracy, although interventions on some discovered sub-concepts have no effect or very slightly decrease task accuracy. Interventions in HiCEMs trained with sub-concept labels from Concept Splitting tend to increase task accuracy, whereas interventions in HiCEMs without sub-concept supervision (our control) can decrease it, highlighting the value of Concept Splitting. On the CUB and PseudoKitchens

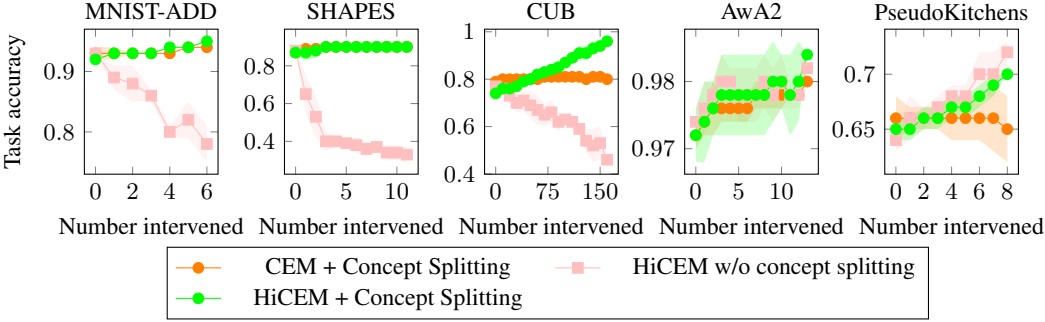

Figure 4: Task accuracy as discovered concepts are intervened. Intervening on discovered sub-concepts improves task accuracy. In some cases, such as in CUB and PseudoKitchens, interventions in HiCEMs lead to a greater increase in task accuracy than in CEMs trained with Concept Splitting's discovered concepts. LF-CBMs (Oikarinen et al., 2023) do not easily support interventions.

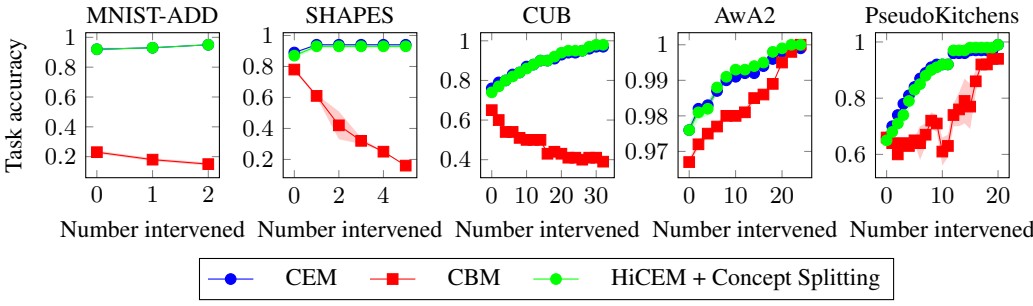

Figure 5: Change in task accuracy as provided concepts are intervened. Provided concept interventions on ImageNet are shown in Appendix I. Provided concept interventions work just as well in HiCEMs as they do in CEMs.

datasets, sub-concept interventions in HiCEMs are more effective than the equivalent interventions in CEMs with both top-level concepts and discovered sub-concepts, supporting the use of the HiCEM architecture over the regular CEM architecture when we have hierarchical concept relationships. HiCEMs with discovered sub-concepts support both the fine-grained interventions enabled by these sub-concepts and the broader concept interventions available in standard CEMs.

## 6 LIMITATIONS AND CONCLUSION

In this work, we introduced Concept Splitting and HiCEMs to enable the discovery and modelling of hierarchical sub-concepts in interpretable models. Our experiments, which use our new PseudoKitchens dataset and include a user study, show that Concept Splitting discovers interpretable sub-concepts. HiCEMs incorporating these concepts provide fine-grained explanations while requiring few manual concept annotations, without sacrificing task or provided concept accuracy. After they have been introduced, discovered sub-concepts can be predicted with high accuracy, and a human expert can correct sub-concept mispredictions. One limitation of our approach is that SAEs are not guaranteed to discover meaningful concepts. Additionally, we only looked at modelling sub-concept relationships. Future work could explore extending our approach to encompass deeper hierarchies. It would also be valuable to investigate, perhaps through a user study, whether offering more fine-grained intervention options is beneficial to a user's cognitive load when interacting with the model. Nonetheless, capturing the hierarchical structure present in the concept embeddings of CEMs helps to fill a gap in earlier concept-based architectures and represents a meaningful step forward in concept-based interpretable modelling.

## ETHICS STATEMENT

Our research aims to enhance the transparency and accountability of neural networks. We have taken care to address ethical considerations related to our experiments. The user study (Appendix F) was conducted under institutional ethics approval, with all participants providing informed consent and their data being fully anonymised. Our new PseudoKitchens dataset is entirely synthetic, containing no personally identifiable information, and all real-world datasets are public benchmarks used in accordance with their licenses.

## REPRODUCIBILITY STATEMENT

We have made a comprehensive effort to ensure the reproducibility of our results. Our source code, which includes implementations of our HiCEM architecture and the Concept Splitting method, has been released in a MIT-licensed public repository.[1] The architectural details of HiCEMs are described in Section 4, and our Concept Splitting method is detailed in Section 3, with an alternative clustering-based method in Appendix A. Experimental settings are specified in Appendices D and E. Details regarding our new PseudoKitchens dataset are in Appendix B. Finally, the methodology for our user study is provided in Appendix F.

### ACKNOWLEDGMENTS

This work was supported by the Engineering and Physical Sciences Research Council [EP/Y030826/1]. A significant portion of this work was carried out whilst MEZ was at the University of Cambridge, funded by the Gates Cambridge Trust via a Gates Cambridge Scholarship.

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

# A SPLITTING CONCEPTS USING CLUSTERING

To investigate other methods for extracting sub-concepts from a CEM's embedding space, we explored an alternative approach based on unsupervised clustering. While the main paper focuses on the SAE-based method, we present the clustering-based alternative here for completeness. Unlike the feature-based discovery of SAEs, where a single input can activate multiple features, this clustering method naturally produces groups of mutually exclusive sub-concepts.

Specifically, we make use of the TURTLE framework (Gadetsky et al., 2024), which is designed to discover the labels of a dataset without any supervision by finding the labelling that induces maximal margin classifiers in the representation spaces of different foundation models. TURTLE can be understood as a method to cluster examples using embeddings from multiple models simultaneously. This is useful in our setup, as we can leverage initial CEMs trained with different backbones (e.g., CLIP (Radford et al., 2021) and DINOv2 (Oquab et al., 2024)) to produce a more robust clustering.

The process begins by partitioning the training data's concept embeddings. For a given top-level concept $c_i$, we create two sets of embeddings: one set containing embeddings from examples where the initial CEMs agree $c_i$ is present, and another where they agree it is absent. We then run the TURTLE clustering algorithm on each of these sets independently. This separation allows us to discover "positive" sub-concepts (which are only active when $c_i$ is present) and "negative" sub-concepts (which are only active when $c_i$ is absent).

To determine the optimal number of sub-concepts to discover for a given set of embeddings, we test a range of cluster counts (from a minimum $\alpha$ to a maximum $\beta$). For each count, we compute the average silhouette score (Rousseeuw, 1987) of the resulting clustering. The number of clusters that yields the highest silhouette score is selected. Once the optimal clustering is found, each cluster is treated as a distinct discovered sub-concept. A key property of this approach is that the clustering algorithm creates a hard partition of the embedding space, meaning each example can belong to only one cluster. Consequently, the discovered sub-concepts (within either the positive or negative set) are inherently mutually exclusive. We generate new binary concept labels for each cluster, where examples belonging to the cluster are labelled as having the sub-concept, and all other examples are labelled as not having it.

Our complete clustering-based Concept Splitting method is detailed in Algorithm 1.

---

**Algorithm 1:** Concept Splitting using clustering for a single concept $c$.

---

**Input:** Set of examples $D$ where all the initial CEMs $M$ agree that the concept we are splitting $c$ is present (or they all agree $c$ is not present), and concept embeddings $Z$ for $c$ from the models in $M$ for all examples in $D$.
**Output:** Discovered concept labels $L$.
**Hyperparameters:** Minimum and maximum number of clusters, $\alpha, \beta \in \mathbb{N}$.
**Note:** TURTLE refers to the TURTLE clustering method proposed by Gadetsky et al. (2024).

1 $L \leftarrow \varnothing$
2 $n \leftarrow \arg\max_{\alpha \leq i \leq \beta}(\texttt{SilhouetteScore}(\texttt{TURTLE}(Z, i)))$
3 $\texttt{clusters} \leftarrow \texttt{TURTLE}(Z, n)$
4 **for** $\texttt{cluster}$ **in** $\texttt{clusters}$ **do**
5    $\texttt{new\_concept\_labels} \leftarrow \texttt{on}$ for all examples in $\texttt{cluster}$ and $\texttt{off}$ for the remaining examples in the training dataset.
6    $L \leftarrow L \cup \{\texttt{new\_concept\_labels}\}$
7 **end**
8 **return** $L$

---

## A.1 EVALUATING AND NAMING DISCOVERED SUB-CONCEPTS

To quantitatively evaluate the interpretability of the sub-concepts discovered via clustering, we automatically assign a human-understandable meaning to each one. This is achieved by matching them against a predefined "concept bank" of ground-truth concepts that were intentionally excluded from the initial CEM training.

Table 5: Mean ROC-AUC for discovered concepts. The clustering method performs better in some cases, and the SAE method performs better in others.

| | MNIST-ADD | SHAPES | CUB | AwA2 | PseudoKitchens |
|---|---|---|---|---|---|
| HiCEM + Concept Splitting (clustering) | $\mathbf{0.94}_{\pm\mathbf{0.02}}$ | $\mathbf{0.94}_{\pm\mathbf{0.03}}$ | $\mathbf{0.90}_{\pm\mathbf{0.01}}$ | $0.81_{\pm0.03}$ | $0.86_{\pm0.03}$ |
| HiCEM + Concept Splitting (SAE) | $0.93_{\pm0.01}$ | $0.93_{\pm0.01}$ | $0.85_{\pm0.01}$ | $\mathbf{0.88}_{\pm\mathbf{0.01}}$ | $\mathbf{0.88}_{\pm\mathbf{0.00}}$ |

Table 6: Task accuracies. The clustering method and the SAE method perform similarly.

| | MNIST-ADD | SHAPES | CUB | AwA2 | PseudoKitchens |
|---|---|---|---|---|---|
| HiCEM + Concept Splitting (clustering) | $\mathbf{0.93}_{\pm\mathbf{0.00}}$ | $\mathbf{0.88}_{\pm\mathbf{0.02}}$ | $\mathbf{0.76}_{\pm\mathbf{0.00}}$ | $\mathbf{0.98}_{\pm\mathbf{0.00}}$ | $0.63_{\pm0.00}$ |
| HiCEM + Concept Splitting (SAE) | $0.92_{\pm0.00}$ | $0.87_{\pm0.02}$ | $0.74_{\pm0.01}$ | $\mathbf{0.98}_{\pm\mathbf{0.00}}$ | $\mathbf{0.65}_{\pm\mathbf{0.01}}$ |

Table 7: Mean ROC-AUCs for provided concepts. The clustering and SAE methods perform identically.

| | MNIST-ADD | SHAPES | CUB | AwA2 | PseudoKitchens |
|---|---|---|---|---|---|
| HiCEM + Concept Splitting (clustering) | $\mathbf{0.99}_{\pm\mathbf{0.00}}$ | $\mathbf{1.00}_{\pm\mathbf{0.00}}$ | $\mathbf{0.93}_{\pm\mathbf{0.00}}$ | $\mathbf{1.00}_{\pm\mathbf{0.00}}$ | $\mathbf{0.91}_{\pm\mathbf{0.00}}$ |
| HiCEM + Concept Splitting (SAE) | $\mathbf{0.99}_{\pm\mathbf{0.00}}$ | $\mathbf{1.00}_{\pm\mathbf{0.00}}$ | $\mathbf{0.93}_{\pm\mathbf{0.01}}$ | $\mathbf{1.00}_{\pm\mathbf{0.00}}$ | $\mathbf{0.91}_{\pm\mathbf{0.00}}$ |

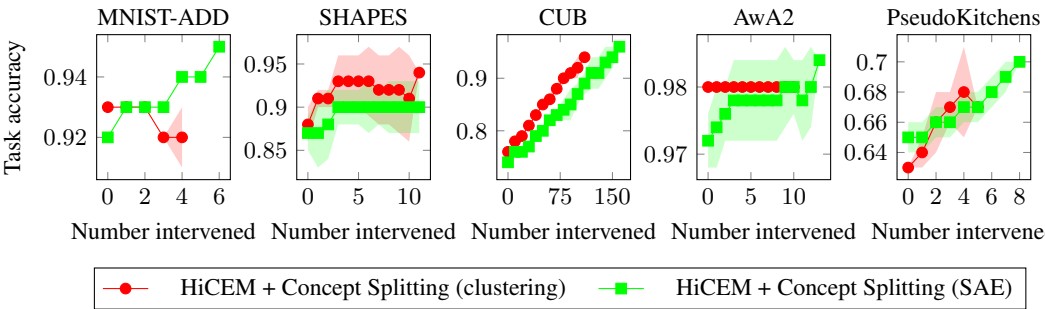

Figure 6: Task accuracy as discovered concepts are intervened. Interventions are mostly effective on concepts discovered with clustering and with SAEs.

The matching methodology used for this clustering-based approach differs from the one used for the SAE-based method in the main paper. Here, for each discovered sub-concept (i.e., each cluster), we compute its ROC-AUC score against every compatible concept within the concept bank. The discovered sub-concept is then assigned the semantic label of the bank concept that yields the highest ROC-AUC score. This procedure ensures that every discovered cluster is assigned an interpretation. A consequence of this approach is that multiple discovered sub-concepts may be matched to the same ground-truth concept from the bank. In our analysis, we treat such instances as duplicates and merge them into a single, final sub-concept before reporting accuracies and performing interventions.

Using clustering to discover sub-concepts is compared to our SAE-based method in Tables 5, 6 and 7 and Figures 6 and 7. Both methods can discover sub-concepts, but the SAE method is less computationally demanding, as it does not require repeated clustering to find a good number of clusters. Additionally, the SAE method does not require a deduplication step. Therefore, it has several advantages over the clustering method.

# B PSEUDOKITCHENS

This Appendix describes PseudoKitchens, our synthetic dataset of photorealistic 3D kitchen renders with ground-truth concept annotations.

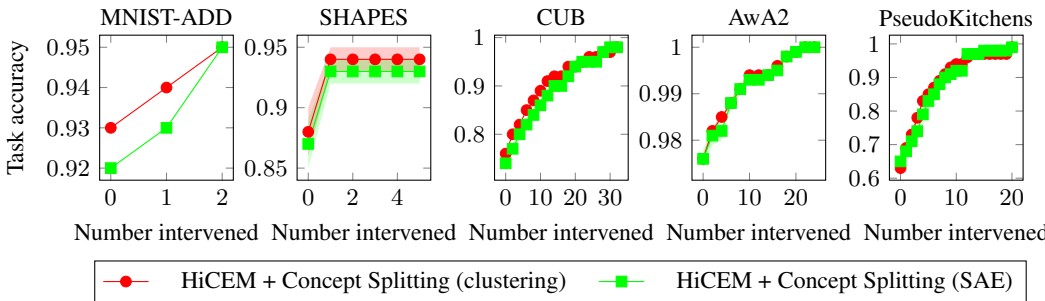

Figure 7: Change in task accuracy as provided concepts are intervened. The two methods perform similarly.

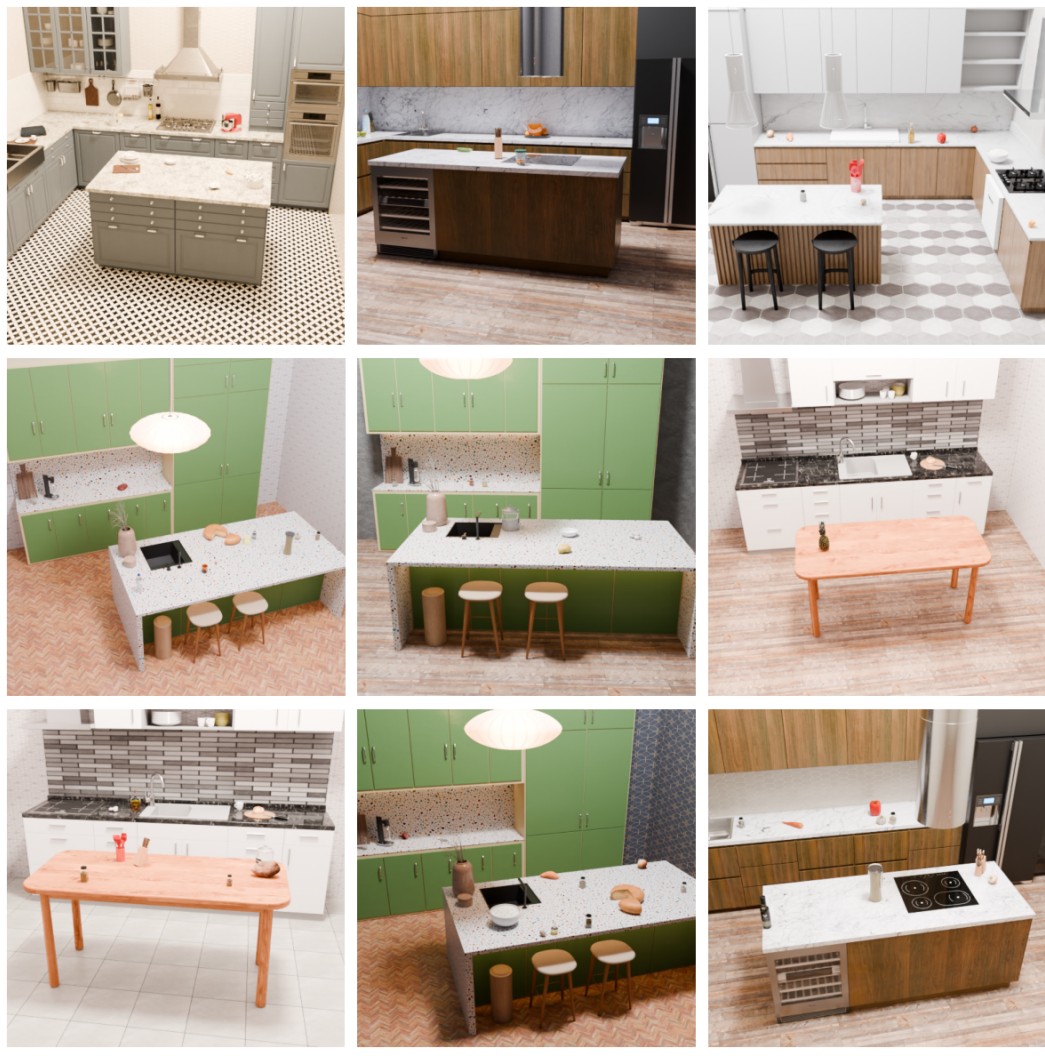

Figure 8: Images from PseudoKitchens demonstrating the dataset's photorealistic quality and diversity in kitchen layouts, ingredient combinations, lighting conditions, and camera perspectives. Each scene contains ingredients for recipe classification.

PseudoKitchens (Figure 8) is generated using Blender 4.5[2], a professional open source 3D graphics software package. We use Blender's Python API to automate scene generation. Our approach

---

[2]https://www.blender.org

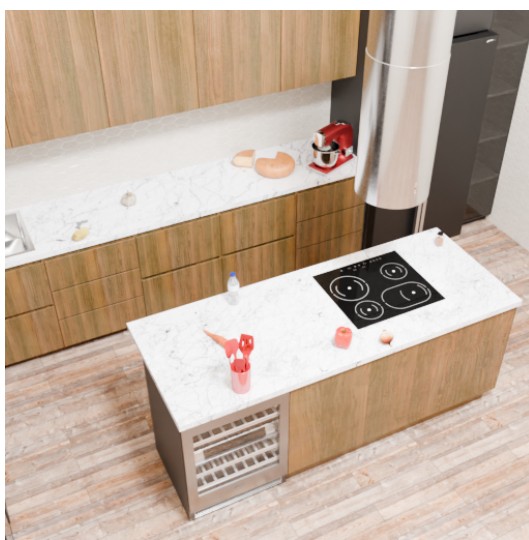 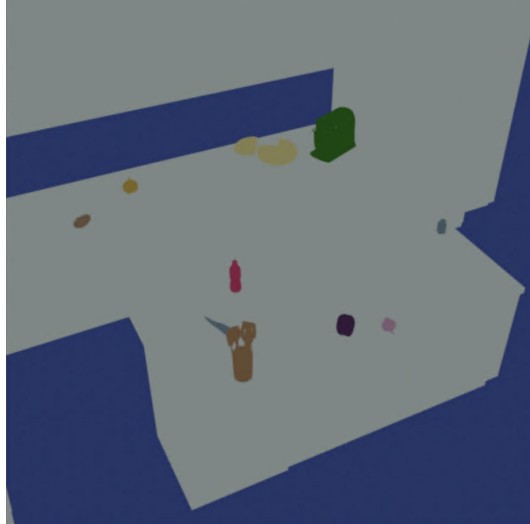

Figure 9: An example from the PseudoKitchens dataset. Left: a photorealistic kitchen scene. Right: ground-truth concept annotations, with colours indicating the spatial locations of individual concepts.

leverages physically-based rendering to create photorealistic images whilst maintaining complete experimental control over scene properties. The dataset consists of kitchen scenes containing ingredients for recipe classification tasks, with each scene accompanied by annotations that describe the location of every ingredient in it.

### B.1 3D ASSETS

Kitchen environments are constructed using 3D assets sourced from BlenderKit[3], all licensed under Royalty Free or Creative Commons CC0 licences. The base kitchen layouts feature countertops, cabinets, appliances, and storage areas. We manually curated five distinct kitchen layouts.

### B.2 RECIPES

We designed 10 distinct recipes that define valid combinations of ingredients for the classification task. Some ingredients are organised into groups as shown in Table 8. If a recipe contains an ingredient group, a random number of ingredients are selected from that group, unless the group is pasta in which case only one type of pasta is selected. The recipes used are shown in Table 9. All together, the recipes use 29 different ingredients. Where possible, for each ingredient we found multiple distinct 3D models to provide variation. For each instance, a recipe is chosen uniformly at random.

Table 8: Ingredient groups in PseudoKitchens.

| Group | Ingredients |
| --- | --- |
| Fruit | Banana, Orange, Apple, Pear, Pineapple |
| Vegetables | Onion, Carrot, Potato, Pepper, Courgette |
| Pasta | Macaroni, Spaghetti |

### B.3 INSTANCE GENERATION

For each generated image:

1. A kitchen layout, floor and wall textures are selected uniformly at random. The light position, intensity and colour temperature are chosen randomly.

---

[3]https://www.blenderkit.com

Table 9: Recipes in PseudoKitchens.

| Recipe | Ingredients |
|---|---|
| Fruit Salad | Fruit |
| Vegetable Pasta | Pasta, Onion, Garlic, Oil, Vegetables, Spice, Tin Tomatoes |
| Risotto | Cheese, Onion, Garlic, Vegetables, Oil, Spice, Rice |
| Chips | Potato, Oil, Flour, Garlic, Spice |
| Chilli | Mince, Oil, Onion, Garlic, Chilli, Tin Tomatoes, Spice, Rice |
| Smoothie | Milk, Yoghurt, Fruit |
| Hot Chocolate | Chocolate, Milk |
| Banana Bread | Butter, Sugar, Egg, Flour, Banana |
| Chocolate Fudge Cake | Egg, Sugar, Oil, Flour, Chocolate, Syrup, Milk |
| Carbonara | Garlic, Meat, Butter, Cheese, Egg, Spaghetti, Spice |

2. The camera viewpoint is randomised within predefined bounds for each kitchen, varying both angle and distance to ensure diverse perspectives whilst maintaining ingredient visibility. In some cases, not all of the ingredients placed will be visible. This mirrors real images, where some features might be occluded or out of the shot. The spatial concept annotations (Figure 9) describe exactly which ingredients are visible in each image.

3. A physics-aware placement system positions ingredients on available surfaces (countertops or tables) using weighted random selection based on surface area. Objects are not allowed to overlap, and are randomly rotated and scaled to provide variation. Task-irrelevant objects, such as saucepans and cooking utensils, are randomly placed in scenes.

### B.4 GROUND-TRUTH ANNOTATIONS

A key advantage of our synthetic approach is the automatic generation of perfect ground-truth annotations. For each rendered scene, we provide:

1. **Concept Location Annotations:** Using Blender's Cryptomatte[4] support, we produce pixel-perfect segmentation masks for every ingredient placed in every image. An example is shown in Figure 9.

2. **Instance Information:** For every image in the dataset, we save a JSON file containing all the information needed to recreate it. This includes the names of all the objects in the image, along with complete scene parameters including camera position, lighting conditions, material assignments, and object transformations. This enables reproducible generation and systematic manipulation for controlled experiments.

### B.5 DATASET COMPOSITION

The complete PseudoKitchens dataset comprises 10,000 training images, 1,000 validation images, and 1,000 test images. Each image is rendered at $512 \times 512$ resolution using the Cycles[5] path tracing renderer, providing photorealistic images whilst maintaining reasonable computational requirements. It takes approximately 10 seconds to render one image on an NVIDIA GeForce RTX 4090 GPU.

## C DATASETS

### C.1 MNIST-ADD

Examples in MNIST-ADD contain two handwritten digits from the MNIST dataset (LeCun et al., 2010), each between 0 and 6 (inclusive). The label is the sum of the digits (so there are 13 classes). There are two provided concepts: the first one indicates whether the first digit is greater than three, and the second one indicates whether the second digit is greater than three. The concepts in the

---

[4] https://github.com/Psyop/Cryptomatte
[5] https://www.cycles-renderer.org

Table 10: High-level concepts and their corresponding merged fine-grained concepts in the AwA2 task.

| High-level concept | Merged concepts |
|---|---|
| patterned | patches, spots, stripes |
| distal_limb | flippers, hands, hooves, pads, paws |
| teeth | chewteeth, meatteeth, buckteeth, strainteeth |
| weapons | horns, claws, tusks |

concept bank consist of a one-hot representation of the digits (for example, one of the concepts is "the first digit is 4"). The dataset includes 10,000 training, 2,000 validation, and 10,000 test samples.

## C.2 SHAPES

Images in our SHAPES task (inspired by the dSprites dataset (Matthey et al., 2017)) contain either a square, circle, triangle or hexagon. The shape and the background are of different colour, and can be red, green, blue or purple. The label of a sample encodes the shape, its colour and the background colour. There are 48 classes. Images are covered in small black polygons to make the task harder. The provided concepts are "the shape is a polygon", "the shape has a light colour", "the shape has a dark colour", "the background is light" and "the background is dark". The concepts in the concept bank form one-hot representations of the shape, its colour and the background colour (for example one of the concepts is "the shape is a square"). The dataset includes 10,000 training, 2,000 validation, and 10,000 test samples. The code we used to generate the dataset is included in the supplementary material, licensed under the MIT License.

## C.3 CUB

CUB (Wah et al., 2011) contains images of birds. Each image is labelled with the species of the bird it contains, along with many concept annotations. There are 200 different species of bird in the dataset. We copy the concept preprocessing performed by Koh et al. (Koh et al., 2020), except we do not filter out any concepts. Some of the concepts in the CUB dataset encode the colour of various parts of the bird. For each bird part $b$ that has concepts indicating its colour, our task contains two provided concepts: "$b$ has a light colour" and "$b$ has a dark colour". This leaves us with 32 provided concepts. The concept bank contains concepts corresponding to the actual colour of the bird parts. The code we use to process the concepts is included in the supplementary material.

## C.4 AwA2

The AwA2 dataset (Xian et al., 2019) comprises images of 50 animal classes, each annotated with semantic concepts. For our task, we define four high-level concepts: *patterned*, *distal_limb*, *teeth*, and *weapons*. Each of these is constructed by grouping related fine-grained concepts, as detailed in Table 10. The concept bank contains the fine-grained concepts. The initial CEM's concepts are the four high-level concepts, as well as the AwA2 concepts that are not used to construct the high-level concepts. We only split the four high-level concepts, as these are the concepts for which we have sub-concepts in the concept bank. The AwA2 image data was collected from public sources, such as Flickr, in 2016 (Lampert et al., 2017). The dataset curators ensured that only images licensed for free use and redistribution were included.

## C.5 PSEUDOKITCHENS

PseudoKitchens is described in detail in Appendix B. The concepts provided to the initial CEM are the ingredient groups in Table 8 (e.g., "contains fruit"), as well as all the ingredients that are not part of a group. The concept bank concepts correspond to the ingredients in the ingredient groups (e.g., "contains apples").

## C.6 IMAGENET

ImageNet (Russakovsky et al., 2015) (the ImageNet Large Scale Visual Recognition Challenge 2012-2017 image classification and localization dataset) spans 1,000 object classes (organised according to the WordNet hierarchy) and contains 1,281,167 training images, 50,000 validation images and 100,000 test images. As labels for the test images are not publicly available, we use the validation images as our test set and split the training images into a training set (1,231,167 images) and a validation set (50,000 images).

The concept labels provided to the initial CEM are generated automatically using the WordNet hierarchy underlying ImageNet. Each ImageNet class is mapped to its WordNet synset, and we collect all of its hypernyms (ancestor categories). A fixed set of 55 high-level concept synsets (e.g. plant, tool, vehicle) is then checked against these hypernyms, and all images from a class are labelled with every concept it descends from. We do not construct a concept bank for the ImageNet task.

## D   LF-CBM AND PCBM BASELINES

**Label-free CBM baseline**   Label-free CBMs (LF-CBMs) discover concepts by prompting large language models for concept names, and then using multimodal models to align LF-CBM representations with these concepts (Oikarinen et al., 2023). Evaluating the accuracy of the discovered concepts requires access to ground truth labels. Since Oikarinen et al. (Oikarinen et al., 2023) evaluate LF-CBMs on the CUB dataset, we use the concepts they discovered and manually align them with our ground truth labels for evaluation. The resulting matchings are shown in Table 11. For our other datasets, we skip the language model prompting step and directly use the names of labelled concepts to train LF-CBMs. This approach was not successful on MNIST-ADD or PseudoKitchens, so we exclude those results. As Oikarinen et al. (2023) provide limited discussion of concept interventions, we do not attempt them in our experiments. Throughout, we use the same hyperparameters as those in the authors' released code for the CUB dataset (Oikarinen, 2023).

**Post-hoc CBM baseline**   Post-hoc CBMs (PCBMs) project the embeddings from a backbone network onto a concept subspace defined by a set of concept vectors (Yuksekgonul et al., 2023). An interpretable predictor then classifies examples based on these projections. To improve predictive performance, a hybrid variant (PCBM-h) includes an additional residual predictor alongside the concept-based one. When concept annotations are available—even if only for part of the training data—they can be used to compute concept vectors (Kim et al., 2018). Otherwise, we can leverage multimodal models by encoding concept names with a text encoder to obtain the vectors. In our experiments, we use this approach for concepts where we have ground truth labels, which we use to evaluate concept accuracy. For all runs, we adopt the same hyperparameters as those used in the authors' released code (Yuksekgonul, 2022).

## E   MODEL ARCHITECTURES, TRAINING AND HYPERPARAMETERS

**Model architectures**   We use the CLIP ViT-L/14 foundation model (Radford et al., 2021) as the backbone for all of our models and baselines. We do not fine-tune the foundation model: we just use the representations it outputs. We use the CLIP ViT-B/16 (Radford et al., 2021) multimodal model in the LF-CBM baseline to align the models' representations with concepts (although the backbone of the LF-CBMs is the CLIP ViT-L/14 model). For all the models, we precompute the representations with standard image preprocessing pipelines and do not use any data augmentations.

When training BatchTopK SAEs, we use the default hyperparameters in the code released by Bussmann et al. (2024) for all datasets.[6] The key distinction of this method is its enforcement of sparsity: it selects the top $n \cdot k$ activations across an entire batch of $n$ samples. This allows the number of active features to vary per sample, targeting an average of $k = 32$ active features. The SAEs are trained for 300 epochs with a dictionary size of 12,288 and a learning rate of $3 \times 10^{-4}$.

When performing Concept Splitting with clustering (Appendix A), we use concept embeddings from two CEMs to cluster examples. One of the CEMs has the DINOv2 ViT-g/14 foundation model

---

[6]https://github.com/bartbussmann/BatchTopK/blob/main/config.py

Table 11: Matches between concepts discovered by the LF-CBM baseline and ground truth concepts in the CUB dataset.

| Discovered concept | Ground truth concept |
|---|---|
| A red eye | has_eye_color::red |
| Glossy black wings | has_wing_color::black |
| a black back | has_back_color::black |
| a black beak | has_bill_color::black |
| a black breast | has_breast_color::black |
| a black throat | has_throat_color::black |
| a bright orange breast | has_breast_color::orange |
| a brownish back | has_back_color::brown |
| a greenish back | has_back_color::green |
| a grey back | has_back_color::grey |
| a large bill | has_bill_length::longer_than_head |
| a red belly | has_belly_color::red |
| a red breast | has_breast_color::red |
| a red throat | has_throat_color::red |
| a streaked back | has_back_pattern::striped |
| a streaked breast | has_breast_pattern::striped |
| a white belly | has_belly_color::white |
| a white breast | has_breast_color::white |
| a white underside | has_underparts_color::white |
| a yellow beak | has_bill_color::yellow |
| a yellow crown | has_crown_color::yellow |
| a yellow eye | has_eye_color::yellow |
| all black coloration | has_primary_color::black |
| black eyes | has_eye_color::black |
| blue upperparts | has_upperparts_color::blue |
| blue wings | has_wing_color::blue |
| a duck-like bird | has_shape::duck-like |

(Oquab et al., 2024) as its backbone, and the other uses the CLIP ViT-L/14 model (Radford et al., 2021). When clustering, we run the TURTLE method (Gadetsky et al., 2024) for 1000 epochs, with the warm-start hyperparameter set to true. We set the $\gamma$ hyperparameter to 10 as suggested by Gadetsky et al. (2024).

All of our CEM and HiCEM (sub-)concept embeddings have $m = 16$ activations. Across all datasets we always use a single fully connected layer for label predictor $f$.

**Training hyperparameters** Our models are trained using the Adam optimisation algorithm (Kingma & Ba, 2015) with a learning rate of $1 \times 10^{-3}$. They are trained for a maximum of 300 epochs (or 50 epochs for ImageNet, or 25 for the ImageNet HiCEM), and training is stopped if the validation loss does not improve for 75 epochs. We use a batch size of 256.

In all CEMs, HiCEMs and CBMs the weight of the concept loss is set to $\lambda = 10$. Following (Koh et al., 2020), in MNIST-ADD, CUB and PseudoKitchens we use a weighted cross entropy loss for concept prediction to mitigate imbalances in concept labels. In MNIST-ADD, and in the linear probe used to name concepts for the user study, we also use a weighted cross entropy loss for task prediction to mitigate imbalances in task labels.

When training CEMs and HiCEMs, the RandInt (Espinosa Zarlenga et al., 2022) regularisation strategy is used: at training time, concepts are intervened independently at random, with the probability of an intervention being $p_{int} = 0.25$. We choose $p_{int} = 0.25$ because Espinosa Zarlenga et al. (2022) find that it enables effective interventions while giving good performance.

# F    USER STUDY

This appendix provides details on the user study conducted to evaluate the quality of sub-concept labels generated by Concept Splitting on ImageNet.

## F.1    NAMING IMAGENET DISCOVERED SUB-CONCEPTS

The sub-concept names evaluated in the study were generated through an automated process, similar to the one used by Rao et al. (2024). To assign a human-readable name to a discovered sub-concept, we first trained a linear classifier (a probe) on the CLIP ViT-L/14 image embeddings for the ImageNet training dataset. This probe was trained to distinguish between images that belong to the sub-concept and those that do not. After training, we iterated through a vocabulary of 20,000 common English words (following Oikarinen & Weng (2023)). For each word, we computed its text embedding using CLIP's text encoder. This text embedding was then passed as input to the trained linear probe to obtain a score. The word whose embedding received the highest score from the probe was selected as the name for the sub-concept.

## F.2    ETHICAL CONSIDERATIONS

Prior to commencing the user study, an application for ethical review was submitted to our institution's ethics committee. The project received approval before any participant recruitment or data collection began. All participants were provided with a detailed consent form informing them of the study's purpose, the nature of their participation, data handling, and their right to withdraw at any time.

## F.3    PARTICIPANT RECRUITMENT

Twenty participants were recruited via word-of-mouth and snowball sampling. This convenience sample was primarily composed of students and colleagues from the authors' institution and other academic institutions, as well as friends and family of the authors. Participation was voluntary, and no monetary or other incentives were provided. The only requirements for participation were basic visual recognition abilities and access to a computer with an internet connection. No specific domain knowledge was necessary.

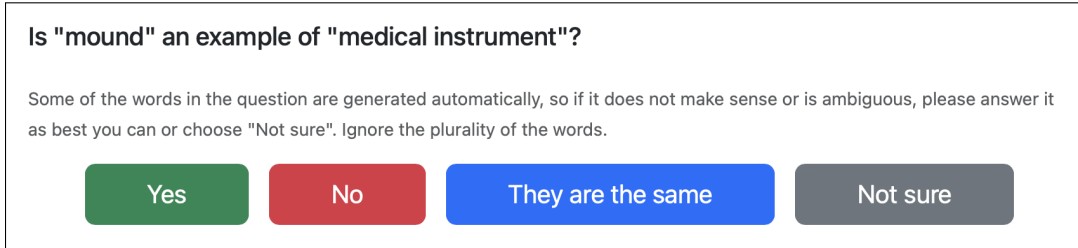

Figure 10: A sub-concept verification question from our user study.

### F.4 STUDY DESIGN AND INTERFACE

The user study was administered through a web-based application, eliminating the need for any software installation on the participants' devices. Upon accessing the study, participants were presented with a consent form. After giving consent, they were assigned a random participant ID to anonymise their responses and allow them to resume the study if they wished.

Participants were then shown a set of instructions detailing the two types of tasks they would be asked to complete. The study consisted of a maximum of 100 questions, and participants could stop at any point.

The two types of questions were:

1. **Sub-Concept Verification:** Participants were asked to evaluate the relationship between a parent concept and a sub-concept by answering the question, "Is [sub-concept] an example of [parent concept]?" (see Figure 10).

   - In the *experimental group*, the sub-concept names provided by our naming method were used.
   - In the *control group*, the sub-concept names were chosen randomly from the dictionary used to name sub-concepts, and paired with a parent concept.

2. **Image Label Verification:** Participants were shown an image and a sub-concept name and asked, "Does this image show [concept]?" (see Figure 11).

   - In the *experimental group*, the images shown were those our method had labelled as exhibiting the given sub-concept.
   - In the *control group*, the images were selected at random from the ImageNet training dataset.

To account for a potential learning curve as participants familiarised themselves with the tasks, the first five responses from each participant for each question type were discarded and not included in our analysis. For each question, participants could also select "They are the same" (for sub-concept verification tasks only) or "Not sure". All participants were shown a mixture of experimental and control questions.

The data was anonymised by assigning a unique, randomly generated ID to each participant, with no personal identifiable information being collected. The collected responses were stored securely and were only accessible to the study's authors.

## G INTERPRETING DISCOVERED CONCEPTS

Figure 12 shows a random sample of 16 images from the MNIST-ADD training dataset that were labelled as having one of the discovered sub-concepts. From this sample, it is clear that the meaning of the discovered sub-concept is "the top digit is 6".

Figure 11: An image label verification question from our user study.

## H    SPLITTING ONE-HOT ENCODED CONCEPT EMBEDDINGS

To explore how Concept Splitting and HiCEMs perform in an idealised setting where concept embeddings contain perfect information about sub-concepts, we conducted a controlled experiment on the MNIST-ADD dataset. Specifically, we created one-hot encoded concept embeddings: for each top-level concept (e.g., the first digit is greater than 3), the positive embedding for that concept was a one-hot vector with a single non-zero entry corresponding to the true sub-concept present in the example (e.g., the first digit is 4). This ensures that the concept embeddings encode perfect, unambiguous information about the sub-concepts that are present. We then applied Concept Splitting to these one-hot embeddings and evaluated whether it could recover the ground-truth sub-concepts.

Concept Splitting with idealised one-hot encoded concept embeddings is compared to Concept Splitting with CEM embeddings in Tables 12, 13 and 14 and Figure 13. When Concept Splitting is applied to concept embeddings that contain perfect (one-hot encoded) information about sub-concepts, it is highly effective, recovering the encoded sub-concepts almost perfectly. In real-world applications, one will not have such ideal concept embeddings. However, as we demonstrate in Section 5, Concept Splitting is also effective when applied to concept embeddings extracted from a CEM.

Table 12: Mean ROC-AUC for discovered concepts. When concept embeddings contain perfect information about sub-concepts, our method effectively recovers them.

|  | MNIST-ADD |
| --- | --- |
| HiCEM + Concept Splitting (one-hot embeddings) | $\mathbf{1.00}_{\pm \mathbf{0.00}}$ |
| HiCEM + Concept Splitting (CEM embeddings) | $0.93_{\pm 0.01}$ |

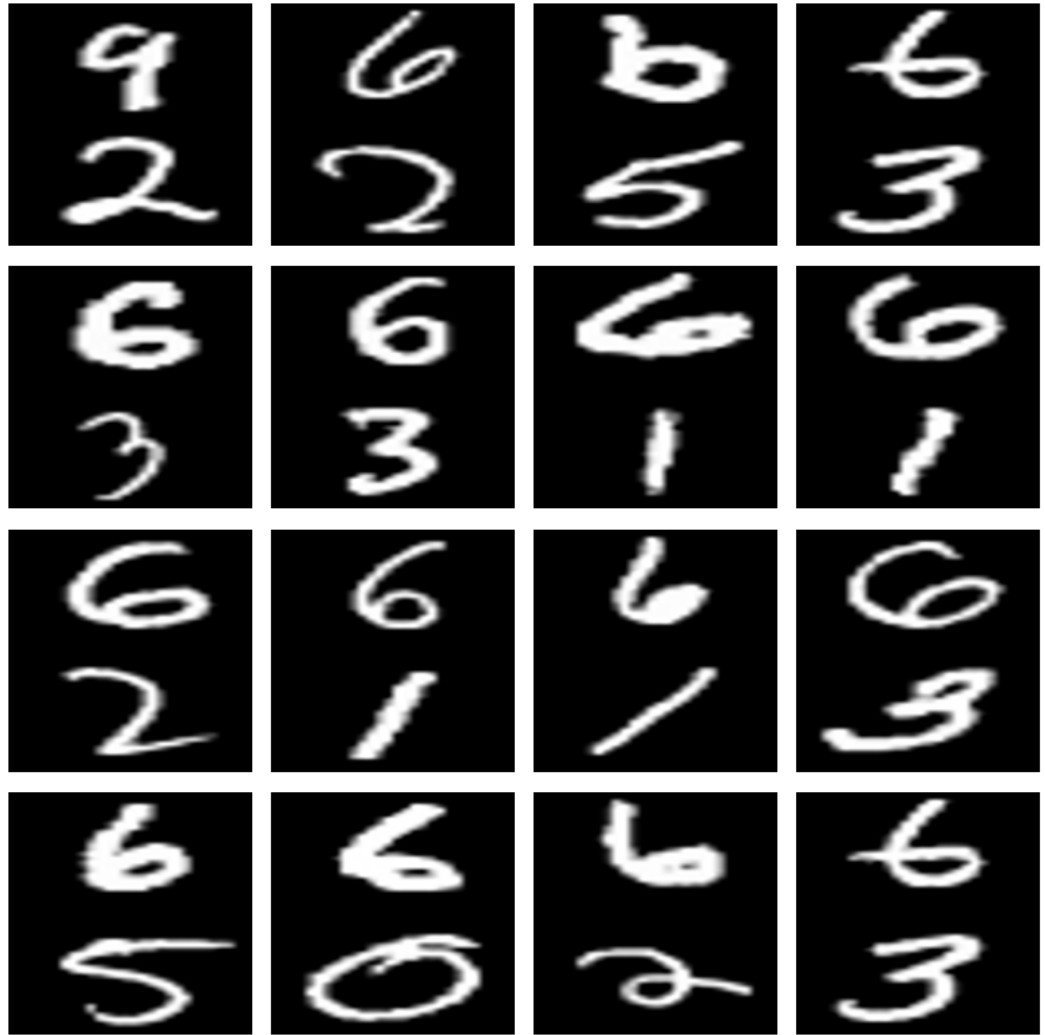

Figure 12: A random sample of images from the MNIST-ADD training dataset that were labelled as having one of the discovered sub-concepts. The interpretation assigned to this concept was "the top digit is 6".

Table 13: Task accuracies. HiCEM achieves similar task accuracy whether trained on concepts discovered from one-hot encoded or CEM embeddings.

|  | MNIST-ADD |
| --- | --- |
| HiCEM + Concept Splitting (one-hot embeddings) | $\mathbf{0.93}_{\pm \mathbf{0.00}}$ |
| HiCEM + Concept Splitting (CEM embeddings) | $0.92_{\pm 0.00}$ |

Table 14: Mean ROC-AUCs for provided concepts. HiCEM achieves similar provided concept accuracy whether trained on concepts discovered from one-hot encoded or CEM embeddings.

|  | MNIST-ADD |
| --- | --- |
| HiCEM + Concept Splitting (one-hot embeddings) | $\mathbf{1.00}_{\pm \mathbf{0.00}}$ |
| HiCEM + Concept Splitting (CEM embeddings) | $0.99_{\pm 0.00}$ |

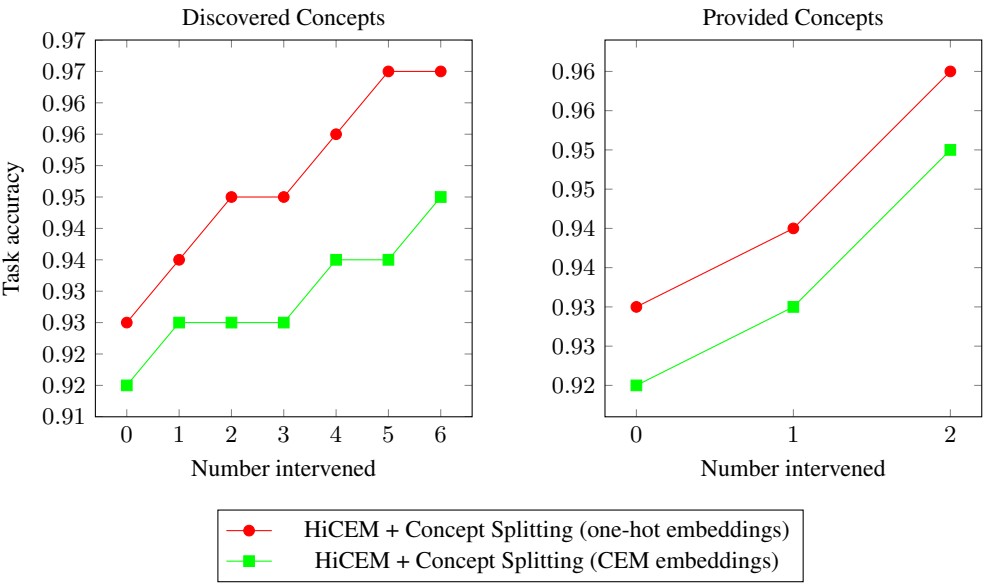

Figure 13: Interventions on discovered concepts are slightly more effective with idealised one-hot encoded embeddings, whilst provided concept interventions perform similarly in both scenarios. Mean and standard deviation over three runs are shown; the standard deviation is negligibly small.

# I IMAGENET CONCEPT INTERVENTIONS

As shown in Figure 14, provided concept interventions on ImageNet perform similarly in the initial CEM and in the HiCEM with discovered sub-concepts. Due to the size of ImageNet, the experiment was only run once so Figure 14 does not contain error bars.

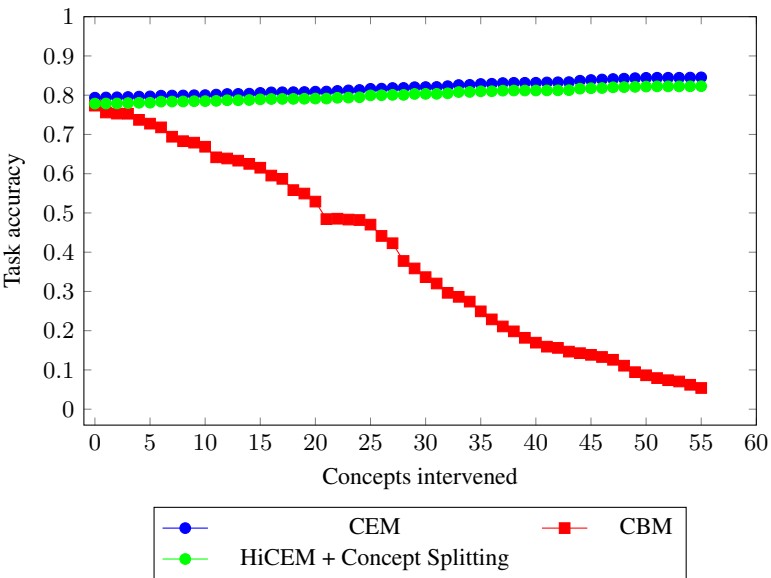

Figure 14: Change in task accuracy as provided concepts are intervened on ImageNet. Provided concept interventions work just as well in HiCEMs as they do in CEMs.

## J  CODE, LICENSES, AND RESOURCES

**Assets**  We used the DINOv2 foundation models (Oquab et al., 2024) (`https://github.com/facebookresearch/dinov2`), whose code and model weights are released under the Apache License 2.0. We also used the CLIP foundation models (Radford et al., 2021) (`https://github.com/openai/CLIP`), whose code is available under the MIT license. To run our experiments, we made use of the CEM (Espinosa Zarlenga et al., 2022) (`https://github.com/mateoespinosa/cem`, MIT license), Post-hoc CBM (Yuksekgonul et al., 2023) (`https://github.com/mertyg/post-hoc-cbm`, MIT license) and Label-free CBM (Oikarinen et al., 2023) (`https://github.com/Trustworthy-ML-Lab/Label-free-CBM`) repositories. We implemented our experiments in Python 3.11 and used open-source libraries such as PyTorch 2.5 (Paszke et al., 2019) (BSD license) and Scikit-learn (Pedregosa et al., 2011) (BSD license). We have released the code required to recreate our experiments in a MIT-licensed public repository.[7]

**Resources**  All of our experiments were run on virtual machines with at least 8 CPU cores, 18GB of RAM, and an NVIDIA GPU (Quadro RTX 8000 or GeForce RTX 4090). Including preliminary experiments, we estimate that approximately 300 GPU hours were required to complete our work.

**Use of AI**  We used Large Language Models (LLMs) as assistants for drafting and improving the clarity and grammar of this manuscript. LLMs were also used to generate boilerplate code. However, all core research ideas, experimental design, and analysis of the results were conducted by the authors.

---

[7]`https://github.com/OscarPi/cem-concept-discovery`

