# OpenReview forum: "Hierarchical Concept-based Interpretable Models"
_ICLR.cc/2026/Conference — ICLR 2026 Poster_

### Official Review · Reviewer_hpPz · 2025-10-17

**Soundness:** 3
**Presentation:** 3
**Contribution:** 3
**Rating:** 6
**Confidence:** 3

**Summary:**

This work introduces multiple novelties to make classification more interpretable. Basing on CEM embeddings, Concept Splitting discovers new, hierarchical subconcepts based on SAEs. Then, HiCEM integrates these sub-concepts into the CEM structure. To evaluate, PseudoKitchen is introduced, a granular dataset with concept control. Also, for imagenet, a user study is conducted to evaluate the meaningfulness of the discovered subconcepts.

**Strengths:**

* This work integrates SAE's concept discovery capabilities into the CEM framework. While there have been attempts at CBM + SAE combinations, the application of SAEs on the CEM embeddings is a neat idea, as these embeddings naturally lend themselves to such an analysis. How to integrate the discovered concepts into CEMs is nontrivial and it seems the authors engineered a way that works nicely.
* In interpretability research, human user studies are the gold standard, but nearly never done. As such, I appreciate the authors' efforts to conduct a user study, whose results have also positively affected my perception of this work. In general, the results are extensive and span many datasets and different axes of evaluation, which support the claims made in the paper.
* Also, the introduction of PseudoKitchens might be a valuable addition to the literature, as it provides realistic, controllable, sample-specific concept control and could help the field get away from class-based concepts.
* I appreciate that reproducible code is provided.

**Weaknesses:**

* My main concern of this work is that it bases on CEMs. I do not consider CEMs truly inherently interpretable, mainly due to their embedding-based design, which has also been demonstrated by [1,2] that show that CEMs contain a lot of leakage. Therefore, any derivative work also has questionable interpretability in my view. For example, the embeddings of the discovered sub-concepts might contain a lot more information than just with regard to their sub-concept. In this line of thought, the added complexity of the sub-embeddings makes the model more opaque, thereby even harder to interpret. There is a trend of arguing that intervention effectiveness implies that there is no (strong) leakage, however, due to the RandInt strategy at training time, this argument does not hold in my opinion. I can be convinced otherwise, if intervention effectiveness was retained without RandInt training for both CEM and HiCEM.
* Understanding the architecture of HiCEM in Sec. 4.1. took a while. I wonder whether it might be easier to understand by replacing the flowtext with more structured formulas or by using a pseudoalgorithm. Also, I just realized there is no Sec. 4.2, and there shouldn't be any single subsections, but that's unimportant here.
* The paper reiterates that related work does not model concept relationships. I am referring to the sentences "Furthermore, they do not consider the relationships between concepts; instead, they treat all concepts as independent variables" and "However, the modelling of these relationships addresses a gap in previous concept-based architectures, advancing the state of the art in concept-based explainability". In my opinion these statements are misleading. There is a considerable line of literature that tackles precisely this problem [3,4,5,6] and given the focus of this work on modeling these concept relations, this related field should at least be mentioned.

[1] Parisini, Enrico, et al. "Leakage and interpretability in concept-based models." arXiv preprint arXiv:2504.14094 (2025).

[2] Almudévar, Antonio, José Miguel Hernández-Lobato, and Alfonso Ortega. "There Was Never a Bottleneck in Concept Bottleneck Models." arXiv preprint arXiv:2506.04877 (2025).

[3] Havasi, Marton, Sonali Parbhoo, and Finale Doshi-Velez. "Addressing leakage in concept bottleneck models." Advances in Neural Information Processing Systems 35 (2022): 23386-23397.

[4] Vandenhirtz, Moritz, et al. "Stochastic concept bottleneck models." Advances in Neural Information Processing Systems 37 (2024): 51787-51810.

[5] Xu, Xinyue, et al. "Energy-Based Concept Bottleneck Models: Unifying Prediction, Concept Intervention, and Probabilistic Interpretations." The Twelfth International Conference on Learning Representations.

[6] Singhi, Nishad, et al. "Improving intervention efficacy via concept realignment in concept bottleneck models." European Conference on Computer Vision. Cham: Springer Nature Switzerland, 2024.

**Questions:**

I invite the authors to address any of the aforementioned weaknesses, albeit I deem it unlikely that it will positively affect my score.

---

> ### Author Response · Authors · 2025-11-21
>
> Dear Reviewer hpPz,
>
> Thank you for your time and for providing a detailed and thoughtful review. We are very grateful for your positive feedback, especially that you found our application of SAEs to CEM embeddings to be a “neat idea”, and that you appreciate our efforts in conducting a “gold standard” user study and introducing the PseudoKitchens dataset. Your critical feedback has been invaluable, and we have made revisions to the paper to address your concerns.
>
> Below, we reply to your feedback. We understand your stance on your score, but we hope to demonstrate that we have taken your points seriously and that the manuscript is stronger as a result.
>
> ### CEMs and Leakage
> We appreciate the opportunity to clarify our perspective and contributions in this context. We acknowledge the important ongoing discussion in the community regarding "leakage" in concept-based models. Our work approaches this from a different angle: rather than trying to eliminate leakage, we aim to make what is encoded (or "leaked") in the embeddings more transparent. Our central hypothesis is that this "leaked" information often contains meaningful, finer-grained sub-concepts. Concept Splitting is our method to uncover and structure this information, thereby improving interpretability by making the model's representations less opaque.
>
> To address your concern that our method is tied to the specific properties of CEM embeddings, we have performed a new experiment, now in Appendix H, as mentioned in our general response. Here, we apply Concept Splitting to "ideal”, one-hot encoded concept embeddings. The results (Table 12) show that our method is highly effective, recovering the encoded sub-concepts almost perfectly. This demonstrates that the principles of Concept Splitting and the HiCEM architecture are general and not exclusively dependent on the embedding structure of CEMs.
>
> Regarding your suggestion to test intervention effectiveness without RandInt, we view the improvement in intervention efficacy from RandInt as a desirable feature. We do not argue that there is no leakage in the concept embeddings of HiCEMs; indeed the reason that embeddings are used rather than scalar concept predictions is to allow additional information to be transmitted. We note that although leakage is undesirable in a lot of instances, it is highly desirable in some other cases, such as in concept-incomplete setups where one does not want to sacrifice too much accuracy in favour of interpretability. As such, we propose a general method that can work in arbitrary embeddings as long as they encode sub-concept information.
> ### Improving the Clarity of the HiCEM Architecture Description
> We apologise that the description of the HiCEM architecture was difficult to follow. As detailed in our general response, we have rewritten Section 4 for clarity. We have broken down the text into more digestible parts and split the original architecture diagram into two simpler, more focused figures (Figure 2 and Figure 3). There is no longer a single subsection. We hope that the new version is significantly easier to understand.
> ### Revising Claims About Related Work
> Thank you for pointing this out and for providing the relevant references. You are correct; our initial statements on the lack of concept relationship modelling in prior work were too strong. We have revised the paper accordingly. As mentioned in our general response, we have added a new paragraph to Section 2 that discusses the literature on modelling concept relations, and we have modified the sentences you highlighted to more accurately claim that our work addresses a gap in modelling hierarchical relationships and discovering them from coarser provided concepts. We are grateful to you for helping us improve the completeness of our related work section.

---

### Official Review · Reviewer_JuNX · 2025-10-27

**Soundness:** 2
**Presentation:** 2
**Contribution:** 3
**Rating:** 6
**Confidence:** 3

**Summary:**

The paper presents Hierarchical Concept Embedding Models (HiCEMs), extended from Concept Embedding Models (CEMs) to represent inter-concept relationships with various granularities. The work also introduces Concept Splitting, a technique for automatically discovering finer-grained subconcepts from pretrained concept embeddings without requiring additional annotations. Together, these methods enhance interpretability, reduce annotation costs, and enable hierarchy concept interventions while archives comparative performance. The paper also introduces a new synthetic dataset, PseudoKitchens, with precise ground-truth concept annotations and spatial localization.

**Strengths:**

1. The HiCEM archives comparable performance comparing to previous methods.
2. The extensive experiments show that the alternative concept splitting method (clustering) also achieves comparable performance to the original design (SAE).
3. The paper proposes an additional synthetic dataset, PseudoKitchens, a controllable scene with precise annotation.

**Weaknesses:**

1. What is the actual benefit of the hierarchical design? Although the experiments show that HiCEM can leverage subconcepts to intervene in the model and achieve better performance on the CUB dataset, in most cases, its performance is comparable to that of CEM.
2. Although the paper includes a user study to evaluate the connections between subconcepts and their parent concepts, it remains unclear whether these subconcepts also contribute to improving the overall interpretability.
3. In the current version, the authors focus only on a two-level hierarchical concept structure. It would be interesting to explore deeper hierarchies and examine the relationships across multiple levels.

**Questions:**

1. It would be helpful if the authors could provide qualitative comparison results between different methods, which would make the improvements in interpretability more intuitive and easier to observe.

---

> ### Author Response · Authors · 2025-11-21
>
> Dear Reviewer JuNX,
>
> Thank you for your review and for your positive feedback on our work. We are glad you highlighted the comparable performance of HiCEMs and the value of our new PseudoKitchens dataset. We appreciate you allowing us to clarify the benefits of our approach.
>
> Below, we reply to your feedback. If you have further questions or concerns, please let us know. Otherwise, we would sincerely appreciate it if you would consider updating your score after considering our replies.
> ### The Benefit of the Hierarchical Design
> This is a crucial point, and we thank you for asking us to clarify it. As we also address in the "Improving Motivation" section of our general response, the primary benefit of the hierarchical design is not to improve task accuracy, but to enable more fine-grained explanations and interventions, and to reduce the burden of concept annotation.
>
> The fact that HiCEMs achieve comparable task accuracy to CEMs is one of our key results: we gain more fine-grained explanations without sacrificing performance. The benefits of interpretability are twofold:
> 1. Finer-Grained Explanations: HiCEMs provide explanations at multiple levels of granularity. Instead of a standard CEM explaining a prediction with a high-level concept like "contains fruit," a HiCEM can provide a more specific explanation, such as "contains apples." This offers a deeper, more intuitive understanding of the model's reasoning. In Appendix G, we show a qualitative example of this, where it is clear that a discovered sub-concept corresponds to the top digit in an image being a "6".
> 2. Finer-Grained Interventions: Users can correct the model's reasoning at a more precise level. If the model misidentifies the fruit in an image, a user can intervene on the specific "apple" sub-concept, which is more targeted than intervening on the broader "fruit" concept.
> ### Exploring Deeper Hierarchies
> We agree that this is a very interesting direction for future work. We have added this to our discussion in Section 6.

---

### Official Review · Reviewer_ga1u · 2025-11-01

**Soundness:** 2
**Presentation:** 3
**Contribution:** 2
**Rating:** 6
**Confidence:** 3

**Summary:**

The authors introduce HiCEMs, a variant of concept embedding models (CEMs) that 1) supports inter-concept (hierarchical) relationships and 2) in addition to the usual supervised concepts, discovers additional sub-concepts in an unsupervised manner using a sparse autoencoder (SAE).  HiCEMs are compared against a number of CBM-like competitors and the interpretability of sub-concepts the SAE discovers is evaluated through a (small scale) user study.

**Strengths:**

- **Novelty**: HiCEMs combine CEMs with concept discovery.  While the two
  ingredients are well known, their combination (and especially the
  rather intricate sub-concept module that is introduced to combine them) look
  novel to me.

- **Quality**:  Using SAEs/clustering for concept discovery in a
  well-structured embedding space (obtained via per-concept supervision) is
  reasonable, but see below. The choice of research questions is good and the
  experimental setup (choice of datasets and competitors) is sound.

- **Clarity**: the text is generally well written, except for the sub-concept
  modules, see below. The related work is relatively complete.

- **Significance**: the idea of pairing SAEs with concept annotations will
  probably inspire follow-up work. This work also partially fills the gap
  between concept-bottleneck models and neuro-symbolic architectures, which is
  interesting.  The new dataset is also a welcome addition.

**Weaknesses:**

While I am generally positive about the paper, I'd like to raise some issues I found.

- **Quality**

    - My main concern is that SAEs do not provide any sort of guarantee.
      Recent works (which may or may not be under submission at ICLR, I have
      not checked) suggest they *can* recover the underlying generative
      concepts provided these are sparse, but in general the jury is still out
      on this, to the best of my knowledge.  Using SAEs is fine, but the
      authors should be careful in not overselling their ability to recover
      good concepts.

  - The interpretability of SAE sub-concepts rests on the user study.  This is
    however quite small scale (20 participantes; by the way, this information
    is important and belongs in the main text; currently it is "hidden" in Appendix E).
    Are these results statistically significant?

  - I also have a (relatively modest) issue with the structure of the user study itself.  The thing is the sub-concepts are named using english words through a "discretization" step (specifically, first english words are mapped to clip embeddings and then hicem sub-concepts via a probe, then sub-concepts are named accordingly). It is possible the sub-concepts don't match the name exactly, but only very approximately, which means that this step essentially "cleans up" the discovered sub-concepts.  To what extent do the results rely on this step? How dissimilar are the names and the associated sub-concepts?

- **Clarity**

  - My main concern is the description of the sub-concept modules, which is
    quite dense and consisting of pure free-form text.  It should be unpacked,
    ideally with the help of equations, to make it clearer and more precise.

  - The CLIP-based automatic naming procedure should be described in the main
    text, as it is integral to the method -- without it, sub-concepts cannot be
    properly interpreted by human stakeholders, breaking self-explainability.

  - I am confused about the term "sub-concept supervision" used in p 7 onward.
    Does it refer to the fact that Concept Splitting is absent? If so, it'd
    make sense to either rename it "sub-concept weak supervision" or to call
    the baseline HiCem-NoSplitting or somesuch.

  - Table 2: "Even better results might be obtained by manually selecting
    high-quality sub-concepts, and naming them manually instead of using our
    crude automatic naming method with CLIP." Very likely, but this is
    equivalent to requiring expert sub-concept supervision - you claim to
    tackle settings in which this is not available.  Moreover, depending on how
    this is obtained (UI-wise), experts could misidentify concepts, so there is
    no a prior guarantee that this would work better than the CLIP probe
    procedure. I think this sentence should be removed.

 - The MNIST-ADD dataset was not introduced by LeCun et al.  Please, fix the
   reference.

 - It should be more clear in the text that the "SHAPES" dataset is similar
   to the relatively well-known dSprites dataset.

**Questions:**

Feel free to comment on any of my observations.

---

> ### Author Response · Authors · 2025-11-21
>
> Dear Reviewer ga1u,
>
> Thank you for your detailed review and positive feedback on our work. We are encouraged that you found the combination of methods to be “novel”, the experimental setup “sound”, and that you believe our work will “probably inspire follow-up work”. We have carefully considered your concerns and have made several revisions to the manuscript to address them.
>
> Below, we reply to your feedback. If you have further questions or concerns, please let us know. Otherwise, we would sincerely appreciate it if you would consider updating your score after considering our replies.
> ### On the Guarantees of SAEs for Concept Discovery
> This is a very fair point. We agree that SAEs are not guaranteed to recover meaningful concepts. However, we note that our core method is not intrinsically tied to SAEs: as we discuss in Appendix A, alternative clustering-based approaches can also be used, demonstrating the modularity of our framework. As concept discovery methods improve in the future, they can be readily integrated into Concept Splitting.
> ### User Study Significance and Naming Procedure
> We agree that the details of the user study are important and should be in the main text. We have moved the number of participants (N=20) into the main body of the paper (Section 5.3). Despite the sample size, the effect size is very large; a Chi-Square test confirms that our results are statistically significant (p < 0.01).
>
> We are not entirely sure we understand your concern regarding the naming step "cleaning up" the discovered sub-concepts. Therefore, we would appreciate it if you could further clarify this point. To describe our process: the discovered sub-concepts are not altered by the naming step. Part of the user study is designed specifically to evaluate the similarity between the assigned name and the images that are labelled as having the sub-concept. If a sub-concept was noisy or incoherent, or the name was a poor match, the user would observe a mismatch between the images and the name, and this would show up in our results. This is precisely why, for our control group, we show random, non-matching name-image pairs to provide a noisy/incoherent matching as a baseline we wish to improve upon.
>
> We do not consider the specific CLIP-based naming procedure to be one of our primary contributions; other methods, such as prototypes [1], could also be used to interpret sub-concepts. In our quantitative evaluation, we use automatic matching to align discovered concepts with held-out concepts in a concept bank for evaluation purposes. We do refer to the CLIP-based naming procedure in Section 5.2, and full details are contained in Appendix F.
> ### Improving the Clarity of the HiCEM Architecture Description
> We appreciate you highlighting the lack of clarity in this section. As noted in our general response, we have rewritten Section 4 based on this feedback. We have broken up the text and split the original, complex diagram into two simpler and more focused figures (Figure 2 and Figure 3). We hope that the architecture is now much easier to understand.
> ### Improved Baseline Naming and Removal of Sentence
> You are right that the term "sub-concept supervision" was confusing. Thank you for the excellent suggestion. We have renamed this baseline to “HiCEM w/o concept splitting” throughout the paper, which is much clearer.
>
> You also correctly pointed out that the sentence in the original Table 2 was speculative and would require expert supervision. We have removed this sentence from the revised manuscript.
> ### Minor Corrections
> Thank you for pointing out the issue with the reference for MNIST-ADD and the lack of context for our SHAPES dataset. We have fixed the reference and have added a sentence clarifying that the SHAPES dataset is inspired by the well-known dSprites dataset. We previously mentioned this in the Datasets appendix, but are happy to clarify in the main body of the paper.
>
> ### References
> [1] D. Alvarez Melis and T. Jaakkola, ‘Towards Robust Interpretability with Self-Explaining Neural Networks’, in Advances in Neural Information Processing Systems, S. Bengio, H. Wallach, H. Larochelle, K. Grauman, N. Cesa-Bianchi, and R. Garnett, Eds, Curran Associates, Inc., 2018.

---

> > ### Comment · Reviewer_ga1u · 2025-11-24
> > **Reply**
> >
> > Thank you for your response.
> >
> > > **We agree that SAEs are not guaranteed to recover meaningful concepts. However, ... clustering-based approaches can also be used***
> >
> > Agreed, but clustering don't provide guarantees either.
> >
> > > **As concept discovery methods improve in the future, they can be readily integrated.**
> >
> > Agreed, but this doesn't mean the approach as-is has any guarantees, or that in the future we will find approaches that provide any.
> >
> > Mind you, I am mildly positive about the paper - the thing I am concerned about is overselling.
> >
> > > **We agree that the details of the user study are important and should be in the main text.**
> >
> > Thank you, this is appreciated.
> >
> > > **We are not entirely sure we understand your concern regarding the naming step "cleaning up" the discovered sub-concepts.**
> >
> > Upon further reflection, I am not sure I understand it fully either. Please ignore this remark.
> >
> > > **We do not consider the specific CLIP-based naming procedure to be one of our primary contributions; other methods, such as prototypes [1], could also be used to interpret sub-concepts.**
> >
> > Yes, and these don't provide any guarantees either.
> >
> > Look, I appreciate all the clarifications and updates, and recall that I am mildly positive about the work. I think that all steps can be justified one way or another. While I do have the feeling that there's something off with the user study, I cannot pinpoint the exact issue, so I am willing to drop my criticism. I also agree that all modules can be hot-swapped for better alternatives if and when those become available. To me this means that I am left with evaluating the idea and the execution. The execution is allright, as we already established. The idea amounts to combining supervised concept learning and unsupervised concept discovery, and its main benefit is that concept supervision can facilitate sub-concept discovery. This is sensible, I would give it a 7/10, which is not an option. I don't feel comfortable upgrading that to 8/10. I hope this doesn't come across as heartless.

---

> > > ### Author Response · Authors · 2025-11-25
> > >
> > > Thank you for taking the time to read and respond to our rebuttal. We have added a sentence to Section 6 (Limitations and Conclusion) to highlight that SAEs are not guaranteed to discover meaningful concepts. We understand your position on your score, and would like to thank you again for your thoughtful comments.

---

### Official Review · Reviewer_5QNW · 2025-11-03

**Soundness:** 3
**Presentation:** 3
**Contribution:** 2
**Rating:** 6
**Confidence:** 4

**Summary:**

This paper introduces two new tools for interpretability specifically targeting the hierarchical nature of concepts/conceptual reasoning. To this end, they introduce a "discovery" interpretability method, as well as a new concept-based deep model to leverage this hierarchical structure. In a nutshell, the discovery method proposed aims to find novel subconcepts from "high-level" concept embeddings using sparse autoencoders. The hierarchical concept embedding model (HiCEM) introduces a new sub-concept neural module that explicitly models a concept starting from its sub-concepts. The authors provide a series of programmatic experiments, as well as human user study to demonstrate their new tools.

**Strengths:**

- *Originality and significance*: while the idea of hierarchical concepts is not new per se, the authors create a novel solution specific for Concept Embedding models which seems compelling.
- *Quality*: overall, the experiments seem reasonable to support the authors claims. In particular, I like the inclusion of user studies and intervention experiments.
- *Clarity*: the paper as a whole is generally understandable.

**Weaknesses:**

- *Originality*: as mentioned above, the idea of hierarchical concepts is not new, e.g. "Hierarchical Concept Discovery Models: A Concept Pyramid Scheme" Panousis 2023. Could the authors expand how their work relates to previous work specifically in the context of hierarchical models of concepts?
- *Significance*: I am completely sold on the idea of concepts being hierarchical (or at the very least not being completely independent). That being said, this does not necessarily mean that modeling them is necessary for better interpretability or intervenability, for the simple reason that often full transparency may be too overwhelming for a person to fully process what a model is doing. Do the authors have in mind a specific real-world use case where their model and concept splitting idea brings a practical advantage compared to other SOTA models?
- *Clarity*: I understand the limitation in space, but I think it would be beneficial to expand on the evaluation of the discovered (sub-)concepts. More below.

**Questions:**

1) Regarding clarity of evaluation of sub-concept discovery:
  - L. 310, do all the provided datasets have concept-subconcepts pairing? If not, how is "between the discovered sub-concept labels and their potential parent-concept-associated matches in the bank." (L.315) evaluated?
  -  Isn't it possible that maybe a (sub-)concept is correctly discovered but for some reason is not in the left-out pool of concepts from the concept bank?
  - How many concepts are left-out? How is the number of features/sub-concepts selected during discovery with SAE? if the number of left out concepts and "features" are not the same, is this taken into account in evaluating the discovery process? I.e. is this considered a multi-label classification problem?
2) How does concept splitting depend on the quality of the top-level concepts? On first thought, I would perhaps think that if the concept embeddings from the pretrained CEM are not good enough, this would hinder the discovery of sub-concepts?
3) From your description, it seems to me that concept splitting does not necessarily depend upon the concepts from CEM. If so, would it then make sense to create some sort of synthetic evaluation experiment, where you run the concept splitting on predefined "perfect" top-level concepts. For example, it shouldn't be a problem to define the concept embeddings as essentially one-hot encoded vectors, I think?
4) As a minor nitpicky suggestion, I would maybe try to expand on why concept splitting is such a pivotal contribution from the authors. Personally, for how it is phrased now, concept splitting sounds like an application of concept discovery methods to pre-trained concept embeddings instead of directly to a latent representation of a sample. So if we put it like this, concept splitting sounds more like a necessary step to then train good HiCEM rather than a key standalone contribution.
5) Based on the authors results, the main advantage of HiCEM vs normal CEMs is essentially the higher intervenability as demonstrated on, eg., the CUB dataset. Could the authors advance hypotheses regarding why this is the case? Is it because CUB and PseudoKitchens are more complex datasets and there actually exists complex parent-subconcept relationships that need direct modeling? Based on this, I believe the limitations section could be expanded by commenting on when it is expected that concept splitting+HiCEM is an extra effort that is worth to do.
6) It would be interesting to have some intervenability human user study to strengthen even more the paper. Something along the lines of "How would you change this explanation to get this prediction?"

---

> ### Author Response · Authors · 2025-11-21
>
> Dear Reviewer 5QNW,
>
> Thank you for your detailed review and constructive feedback. We are very pleased that you are “completely sold on the idea of concepts being hierarchical” and that you found our solution to be “compelling.” Your suggestions for improvement have been invaluable in strengthening the paper.
> ### Relation to Prior Work and Contribution of Concept Splitting
> Thank you for pointing out the need to better situate our work. As mentioned in our general response, we have now expanded Section 2 to include a discussion of "Coarse-to-Fine Concept Bottleneck Models" (the published version of Panousis 2023). Coarse-to-Fine CBMs adopt a spatially-grounded approach to hierarchy (high-level for whole images, low-level for patches) dependent on vision-language models and textual concept descriptions. In contrast, we discover hierarchies by decomposing concept embeddings, independent of spatial structure or vision-language constraints.
>
> Regarding your comment on the contribution of Concept Splitting, we believe that our contribution is showing that CEMs capture sub-concepts not provided during training, and that these sub-concepts can be extracted and used to provide more fine-grained explanations and interventions. This provides a pathway to building hierarchical models without needing to train them from scratch with multi-level annotations.
> ### Practical Advantage and Real-World Use Case
> This is a crucial point, which we have also addressed in the "Improving Motivation" section of our general response. A significant practical advantage of our approach is the **reduction of the concept annotation burden**, which makes concept-based models more viable in the real world. Standard CEMs require labels for every concept they are to model. Obtaining such fine-grained labels can be a significant bottleneck. Our method allows a practitioner to start with a few high-level, coarse-grained concepts, which are much cheaper to annotate, and then use Concept Splitting to automatically discover more specific sub-concepts. We have modified Section 1 to further emphasise this.
>
> The design of the HiCEM architecture means that only relevant sub-concepts (determined by whether or not the parent concept is active) can be predicted. This reduces the number of concepts that users must consider compared to a CEM, which simply models a flat list of concepts. Users do not need to look at sub-concepts if they find them overwhelming.
> ### Evaluation of Discovered Sub-Concepts
> Thank you for the detailed questions on our evaluation setup. To clarify:
> - Our concept banks do contain concept-subconcept pairings. We have clarified this in the revised manuscript (line 357).
> - It is possible for a sub-concept to be correctly discovered but not be present in our concept bank. For our quantitative evaluation, such concepts are not included as we lack ground-truth labels for them. We have clarified this on line 361 of the revised manuscript.
> - The details of the concept banks are described in Appendix C. As described in Appendix E, we use a BatchTopK SAE with a dictionary size of 12,288 and an average of k=32 active features per input across all datasets. The number of features is larger than the number of left-out concepts, so some features are not matched to a concept in the concept bank.
> ### Dependence on Top-Level Concept Quality
> Sub-concept discovery depends on reasonable-quality top-level labels. If such labels are unavailable, they could be formed by grouping semantically related classes, as we do for ImageNet.
> ### Experiment with “Perfect” Concept Embeddings
> This is an excellent suggestion. We agree that concept splitting does not necessarily depend upon using concept embeddings from a CEM, and that an experiment with "perfect" concept embeddings would be highly informative. As detailed in our general response, we have followed your advice and added a **new experiment in Appendix H**. In this idealised setting, we apply Concept Splitting to one-hot encoded vectors that perfectly represent sub-concept information. The results (Table 12) show that our method recovers the encoded sub-concepts almost perfectly.
> ### Advantage of HiCEM vs Regular CEM
> HiCEM models parent-child relationships, ensuring only relevant sub-concepts are predicted and interventions respect the hierarchy.  As you say, in the discovered concept intervention results on the CUB and the PseudoKitchens datasets in Figure 4, the HiCEM outperforms the regular CEM after Concept Splitting. On the other datasets, discovered sub-concept interventions in the regular CEM (after Concept Splitting) are highly effective, not leaving much room for improvement. Across all datasets, however, performing Concept Splitting leads to models that provide finer-grained explanations and opportunities for intervention.
> ### Suggestion for Future Work: Intervenability User Study
> A user study on discovered sub-concept interventions would be valuable. We have added this as future work in Section 6.

---

### Author Response · Authors · 2025-11-21
**General Response to All Reviewers: Key Improvements**

We sincerely thank all reviewers for their time and for providing constructive feedback on our manuscript. We are particularly encouraged that reviewers are “completely sold on the idea of concepts being hierarchical” (5QNW) and highlighted several other strengths, including the work's “originality and significance” (5QNW), its “sound” experimental setup (ga1u), the “extensive experiments” (JuNX), and the user study (hpPz). We are also pleased that reviewers recognised our new PseudoKitchens dataset as a “welcome addition” to the field (ga1u, hpPz). We have found the suggestions to be extremely helpful, and we believe the revised manuscript is substantially stronger as a result.
We have updated the paper, with all major changes highlighted in red for clarity. Below, we summarise the most significant changes, referencing the reviewers whose comments prompted them.


### New Experiment with “Ideal” Concept Embeddings [5QNW, hpPz]
Based on your excellent suggestions, we have added a new experiment in Appendix H to further validate our approach where we run Concept Splitting on “idealised” concept embeddings. These embeddings correspond to one-hot vectors that, for each concept, encode sub-concepts from our concept bank. We find that Concept Splitting almost perfectly recovers the encoded sub-concepts, demonstrating its effectiveness. More importantly, this experiment shows that our method does not necessarily depend on the concept embeddings produced by CEMs and could therefore be applied on top of any concept embedding learning approach.


### Improving the HiCEM Architecture Description [ga1u, hpPz]
Reviewers noted that the description of the HiCEM architecture was dense and difficult to understand. To address this, we have rewritten Section 4 (Hierarchical CEMs) for clarity. We have broken the description of the HiCEM architecture into more digestible parts and split the architecture diagram into two simpler, more focused figures (Figure 2 and Figure 3).


### Strengthening Our Relation to Prior Work [5QNW, hpPz]
We have expanded Section 2 (Background and Related Work) to better situate our contributions within the existing literature.
- [5QNW] We now include and discuss "Coarse-to-Fine Concept Bottleneck Models" (the published version of Panousis 2023), clarifying how our work differs by discovering sub-concepts directly from a pretrained CEM's embedding space without requiring a vision-language backbone.
- [hpPz] We have added a new paragraph on modelling concept relationships to more accurately represent the state of the art and have revised our claims to acknowledge prior work in this area.


### Improving Motivation [5QNW, JuNX]
A key point of feedback from multiple reviewers was the need to better articulate the motivation for our work and its practical advantages. We appreciate this feedback and wish to use this opportunity to reiterate our core motivation. The primary goal of our work is to enhance model interpretability with finer-grained explanations and interventions, rather than to improve task accuracy. Our work directly addresses a key limitation of standard CEMs: the need for exhaustive, fine-grained concept labels during training. The core practical advantage of our method is that it allows practitioners to start with a few coarse-grained labels, which are often cheaper and easier to obtain, and then automatically discover more specific sub-concepts. The coarse-grained labels could be obtained by grouping semantically related classes (as we do for ImageNet in the paper), or perhaps by using LLMs (as seen in label-free methods [1]). This is a central theme of our work, as it makes concept-based models more applicable in real-world scenarios where annotation budgets are limited. We have modified Section 1 (Introduction) to further emphasise this.


### References
[1] Tuomas P. Oikarinen, Subhro Das, Lam M. Nguyen, and Tsui-Wei Weng. Label-free Concept Bottleneck Models. In The Eleventh International Conference on Learning Representations, ICLR 2023, Kigali, Rwanda, May 1-5, 2023. URL https://openreview.net/pdf?id=FlCg47MNvBA

---

### Meta-Review · Area_Chair_ccYp · 2026-01-07

**Summary:**

This paper proposes a hierarchical extension to concept based interpretable model by modeling hierarchical relations between concepts. They further propose an approach to discover finer-grained sub-concepts from already trained CEM models via sparse auto encoders. The contribution is empirical, with results over multiple datasets, a user study, and a new synthetic dataset.

Reviewers agree that the idea to model a hierarchy of concepts - while not completely novel - is neat and timely; the agree that the level of empirical evidence is substantial. The concerns on the reliability of discovery of sub-concepts by the sparse auto encoders is still an open concern that was partially addressed through the provided extra empirical results. Furthermore, the proposed approach can still inherit problems of leakage.

This is a borderline paper, and I'm recommending acceptance.

**Reviewer Concerns:**

### Rev 5QNW

- The reviewer questions the novelty of the work vis a vis prior concept pyramid work. In the rebuttal, the authors extended their related work discussion to include the referenced papers and clarified the differences.

- A question is raised on the practical use of hierarchies in concepts for explanations. The authors argue that a practical advantage can be that of reduced annotations that can be easier to deal with for users/annotators.

- The reviewer had concerns regarding the evaluation for sub-concept discovery, involving the concept bank coverage, the matching procedure, among others. These were clarified by the authors in their rebuttal.

- The reviewer suggested a synthetic case to evaluate a perfect embedding scenario. This was added by the reviewers in the Appendix, showing that their method recovers them accurately.

### Rev ga1u

- They question the ability of the sparse auto encoders to recovery concepts in a meaningful way, and the extent to which the authors were exaggerating these claims. The authors acknowledged this and added limitations to their phrasing.

- They mentioned that the user study is quite small (N=20) and questioned the significance of the results. The authors report a chi square p<0.01 for the user study as a response.

- Other minor suggestions were brought up, addressed in the rebuttal.

### Rev JuNX

- They question on the benefits of the hierarchy in the concepts. The authors stress that the primary goal of the approach is interpretability and interminability, no necessarily accuracy gains.

- They also question the limitation of only 2 levels of hierarchies. This is left as future work by the authors.

### Rev hpPz

- The reviewer argues that CEMs are not inherently interpretable due "leakage", and wonders whether the extra hierarchical structure makes this problem worse. The authors argue that their approach doesn't prevent leakage, but rather provides a picture of what concepts are used (or, potentially leaked).

- The reviewer mentioned that the language used to reference prior work was somewhat misleading, This is acknowledged by the authors and corrected.

**Reviewer Scores:**

- Rev 5QNW (originally 6): Most of the comments raised by the reviewer were addressed. I assume this reviewer would have maintained or increased their scores.
- Rev ga1u (Originally 6): was happy with the rebuttal and the exchanges with the author. Did not want to increase to 8, but was generally supportive.
- Rev JuNX: (6). The reviewer did not engage further, and was only moderately confident (3).
- Rev hpPz (6): Did not engage with authors further. Their comments were mostly addressed, though the concerns were somewhat more philosophical.

---

### Decision · Program_Chairs · 2026-01-26

Accept (Poster)